# Secondary organic aerosol and organic nitrogen yields from the nitrate radical (NO₃) oxidation of alpha-pinene from various RO₂ fates

Kelvin H. Bates,[1,2] Guy J.P. Burke,[1] James D. Cope,[1] and Tran B. Nguyen[1]

[1]Department of Environmental Toxicology, University of California Davis, Davis CA 95616, USA
[2]Center for the Environment, Harvard University, Cambridge, MA 02138, USA

*Correspondence to*: Tran B. Nguyen (tbn@ucdavis.edu)

**Abstract.** The reaction of $\alpha$-pinene with $NO_3$ is an important sink of both $\alpha$-pinene and $NO_3$ at night in regions with mixed biogenic and anthropogenic emissions; however, there is debate on its importance for secondary organic aerosol (SOA) and reactive nitrogen budgets in the atmosphere. Previous experimental studies have generally observed low or zero SOA formation, often due to excessive $[NO_3]$ conditions. Here, we characterize the SOA and organic nitrogen formation from $\alpha$-pinene + $NO_3$ as a function of nitrooxy peroxy ($n RO_2$) radical fates with $HO_2$, $NO$, $NO_3$, and $RO_2$ in an atmospheric chamber. We show that SOA yields are not small when the $n RO_2$ fate distribution in the chamber mimics that in the atmosphere, and the formation of pinene nitrooxy hydroperoxide (PNP) and related organonitrates in the ambient can be reproduced. Nearly all SOA from $\alpha$-pinene + $NO_3$ chemistry derives from the $n RO_2$ + $RO_2$ pathway, which alone has an SOA mass yield of 56($\pm$7)%. Molecular composition analysis shows that particulate nitrates are a large (60-70%) portion of the SOA, and that dimer formation is the primary mechanism of SOA production from $\alpha$-pinene + $NO_3$ under simulated nighttime conditions. Synergistic dimerization between $n RO_2$ and $RO_2$ derived from ozonolysis and OH oxidation also contribute to SOA formation, and should be considered in models. We report a 58 ($\pm$20)% molar yield of PNP from the $n RO_2$ + $HO_2$ pathway. Applying these laboratory constraints to model simulations of summertime conditions observed in the Southeast United States (where 80% of $\alpha$-pinene is lost via $NO_3$ oxidation, leading to 20% $n RO_2$ + $RO_2$ and 45% $n RO_2$ + $HO_2$) , we estimate yields of 11% SOA and 7% particulate nitrate by mass, and 26% PNP by mole, from $\alpha$-pinene + $NO_3$ in the ambient. These results suggest that $\alpha$-pinene + $NO_3$ significantly contributes to the SOA budget, and likely constitutes a major removal pathway of reactive nitrogen from the nighttime boundary layer in mixed biogenic/anthropogenic areas.

# 1 Introduction

Monoterpenes ($C_{10}H_{16}$) are a major class of biogenic hydrocarbons. Although less abundant than isoprene in terms of absolute emission flux of non-methane hydrocarbons, they have a disproportionate importance for the formation of

**Scheme 1:** The $NO_3$-initiated oxidation of $\alpha$-pinene produces a nitrated $RO_2$ radical ($nRO_2$). The major isomer is shown.

secondary organic aerosol (SOA), accounting for half of the total fine aerosol globally (Zhang et al., 2018), and for nitrogen oxide (NO, $NO_2$, $NO_3$) sequestration through the formation of gaseous and particle-phase organic nitrates (Pye et al., 2015). Thus, monoterpene chemistry plays a prevailing role in aerosol-climate interactions and atmospheric air quality. Of the monoterpenes, $\alpha$-pinene is the most abundant globally (Sindelarova et al., 2014). This is especially notable over boreal coniferous forests where the $\alpha$-pinene emission flux alone can overtake isoprene and the combined flux of all other monoterpenes during the summer season (Hakola et al., 2003). The atmospheric abundance, fast reaction rates, and nighttime emission profile of $\alpha$-pinene conspire for it to dominate the fate of the nitrate radical ($NO_3$) in the dark, and also to play a significant role in the daytime (Ayres et al., 2015). The reaction of $\alpha$-pinene + $NO_3$, thus, is one of the most prevalent reactions observed in the summer in mixed biogenic-anthropogenic sites, such as the Southeastern United States.

$NO_3$ reacts with $\alpha$-pinene by addition to the double bond, mainly producing a nitrooxy alkyl radical in the tertiary position, which is rapidly converted to a nitrooxy peroxy radical ($nRO_2$) upon collisions with molecular oxygen (**Scheme 1**). Prominent field observations of a monoterpene nitrooxy hydroperoxide (($O_2NO$)ROOH) further suggest that the $nRO_2$ reacts with $HO_2$ significantly (Nguyen et al., 2015). However, laboratory research on $\alpha$-pinene + $NO_3$ has so far not demonstrated nitrooxy hydroperoxide formation, except for one experiment during the FIXCIT chamber studies where the pinene nitrooxy hydroperoxide (PNP) was abundantly produced from experiments using ppm-level concentrations of formaldehyde to produce dark $HO_2$ via the slow reaction of $CH_2O + NO_3 \rightarrow HO_2 + HNO_3$ (Nguyen et al., 2014a). Although we now appreciate that the $nRO_2$ from $\alpha$-pinene can form monoterpene nitrooxy hydroperoxide, its absolute yield is unknown; furthermore, such high concentrations of formaldehyde in the exploratory FIXCIT experiment will affect SOA formation, and thus, aerosol yields were not extractable.

Determinations of SOA yields of $\alpha$-pinene + $NO_3$ in the literature have also suffered from high uncertainty, although it has generally been accepted that the yields are lower than those of many other monoterpenes and, thus, often not considered for modeling SOA and organic nitrogen formation (Pye et al., 2015; Ayres et al., 2015). Fry et al. (2014) reported an SOA yield of exactly zero from this reaction from experiments using $N_2O_5$ as an $NO_3$ precursor. Hallquist et al. (1999), Moldanova et al. (2000), Bell et al. (2021), and Mutzel et al. (2021) also reported low mass yields (0.3–7%, 0.3–6.9%, 3–11%, and 5.9–6.4% respectively, depending on precursor concentrations) with reactions performed similarly. Spittler et al. (2006) performed the experiment with slow and minute introductions of $N_2O_5$ and reported higher mass yields, although they depend on whether the seed particle chosen was ammonium sulfate (9%) or organic (16%). Nah et al. (2016) and Kurtén et al. (2017) also observed

minimal SOA formation (mass yields of 3.6% and <1%, respectively) using $NO_2 + O_3$ as a source of $NO_3$, and formaldehyde to promote $HO_2$ chemistry. From these observations, it is clear that chamber reaction conditions are highly influential in the observed SOA yields, and that previous studies may have each probed different $nRO_2$ fates. Thus, a systematic investigation of how $nRO_2$ fates dictate reaction outcomes will enable reconciliation of past results and accurate representation of this reaction in atmospheric models. The high initial $NO_3$ concentrations (tens of ppb) used in some previous studies, derived from

the decomposition of $N_2O_5$, will cause $nRO_2 + NO_3$ to dominate in the chamber, when it is negligible in the field ([$NO_3$] is persistently at or below the detection limit of 1 pptv in the rural Southeast United States; Ayres et al., 2015). Kurtén and coworkers further illuminated the role of alkoxy radical scission in SOA formation, and predicted low SOA yields from the $\alpha$-pinene + $NO_3$ reaction when the $RO_2$ radical is reduced to RO via bimolecular reaction with $NO_3$, $RO_2$, and even $HO_2$ (Kurtén et al., 2017).

Given the high relevance of the $\alpha$-pinene + $NO_3$ reaction, it is critical to place tighter constraints on how this reaction contributes to SOA and organic nitrogen in the ambient environment. In light of the emerging appreciation for the importance of $RO_2$ radical fate in designing chamber experiments (Nguyen et al, 2014a; Xu et al., 2019; Boyd et al., 2015; Teng et al., 2017; Crounse et al., 2013), we reinvestigate this reaction to probe the SOA yield and organic nitrate formation from $\alpha$-pinene + $NO_3$ from each relevant $nRO_2$ reaction channel. While a chamber experiment may never truly replicate the field, and ours

certainly are no exception, the $nRO_2$ fate distribution in this work was designed to approach those expected in the ambient nighttime (Ayres et all., 2015; Romer et al., 2018), including any reaction synergies that may occur (Kenseth et al., 2018; Inomata, 2021). Finally, relatively little information is available for the $nRO_2$ compared to their hydroxylated counterparts; this work also constrains the rate coefficients and branching ratios of the $\alpha$-pinene $nRO_2$ through a combination of chamber reactions and modeling. We demonstrate a new $HO_2$ formation route in the dark chamber that does not require carbon inputs,

and thus enables SOA yields to be more accurately measured when probing the $RO_2 + HO_2$ pathway from the $\alpha$-pinene + $NO_3$ reaction.

## 2 Methods

### 2.1 Chamber reactions

      Experiments were performed in a 10 $m^3$ FEP Teflon atmospheric chamber in the dark and at a temperature of ~22 °C,

consistent with the average nighttime temperatures in the Southeastern United States in the summer (Hidy et al., 2014). Experiments were performed dry to reduce uncertainty from variable wall loss corrections, to avoid rapid hydrolysis of tertiary nitrates (Vasquez et al., 2021) and the effects of gas-particle partitioning of $H_2O_2$ and $N_2O_5$, and because relative humidity (RH) is not expected to influence radical reactions. Temperature and RH were monitored continuously by a membrane probe (Vaisala Inc.) calibrated with saturated salt solutions. $NO_x$ and $O_3$ mixing ratios were quantified with commercial

chemiluminescence (Thermo 42i) and photometric (Thermo 49i) analyzers. The chemiluminescence analyzer was calibrated

with a $NO_2$ primary standard (200 ppmv, Air Liquide) diluted to desired concentrations using ultra-high-purity (UHP) $N_2$ and zeroed with UHP $N_2$. The photometric analyzer was cross calibrated for $O_3$, produced by an ozone generator (A2Ozone Inc.) from ultra zero air (Air Liquide), with a Fourier-transform infrared spectrometer (FT-IR, Shimadzu Scientific Inst. Inc., IR-Tracer 100) using a 10 m gas cell. α-Pinene mixing ratios were quantified with a gas chromatograph coupled to a flame ionization detector (GC-FID) using a PLOT-Q column (Agilent Inc.) and a custom pneumatically-injected method. The GC-FID was calibrated with a NIST-traceable 40 ppmv α-pinene primary standard in $N_2$ (Matheson Gas) diluted to several concentrations with UHP $N_2$ using calibrated mass flow controllers (SEC, Horiba Inst. Inc.). The mass flow controllers were calibrated using a primary flow calibrator (AP Buck, Inc.).

**Table 1.** Initial conditions for chamber experiments performed in this work. Controls to subtract SOA yields from background chemistry (not listed here) were conducted for experiments 1-11 and 17-25 using identical conditions but omitting of $NO_2$.

| Expt | Type | pinene (ppb) | $H_2O_2$ (ppm) | $O_3$ (ppb) | $NO_2$ (ppb) | NO (ppb) | $N_2O_5$ (ppb) | Mixing time (min) | seed area ($\mu m^2/cm^3$) | seed mass ($\mu g/m^3$) |
|---|---|---|---|---|---|---|---|---|---|---|
| 1 | $RO_2$ fate | 27 | 0 | 53 | 44 | 0 | 0 | 60 | 645 | 23.6 |
| 2 | $RO_2$ fate | 27 | 0 | 99 | 73 | 0 | 0 | 60 | 662 | 24.3 |
| 3 | $RO_2$ fate | 27 | 4 | 50 | 149 | 0 | 0 | 70 | 645 | 24.3 |
| 4 | $RO_2$ fate | 27 | 8 | 53 | 149 | 0 | 0 | 90 | 631 | 23.2 |
| 5 | $RO_2$ fate | 27 | 3.5 | 96 | 225 | 0 | 0 | 120 | 700 | 25.8 |
| 6 | $RO_2$ fate | 27 | 9 | 39 | 99 | 0 | 0 | 120 | 650 | 25.2 |
| 7 | $RO_2$ fate | 27 | 2 | 43 | 125 | 0 | 0 | 60 | 666 | 25.7 |
| 8 | $RO_2$ fate | 27 | 8 | 103 | 145 | 0 | 0 | 120 | 695 | 27.2 |
| 9 | $RO_2$ fate | 54 | 2 | 45 | 120 | 0 | 0 | 60 | 615 | 24.5 |
| 10 | $RO_2$ fate | 130 | 2 | 43 | 121 | 0 | 0 | 60 | 660 | 26.6 |
| 11 | $RO_2$ fate | 27 | 2 | 98 | 86 | 0 | 0 | 120 | 9300 | 392 |
| 12 | $RO_2$ fate | 27 | 0 | 0 | 0 | 0 | 31 | 0 | 7260 | 252 |
| 13 | $RO_2$ fate | 27 | 0 | 0 | 0 | 0 | 200 | 0 | 7960 | 330 |
| 14 | $RO_2$ fate | 60 | 0 | 0 | 11 | 83 | 110 | 0 | 5330 | 214 |
| 15 | $RO_2$ fate | 60 | 0 | 0 | 240 | 110 | 85 | 0 | 5010 | 221 |
| 16 | $RO_2$ fate | 45 | 0 | 0 | 565 | 65 | 80 | 0 | 4380 | 205 |
| 17 | seed area | 27 | 4 | 102 | 72 | 0 | 0 | 75 | 0 | 0 |
| 18 | seed area | 27 | 4 | 97 | 71 | 0 | 0 | 75 | 73 | 2.79 |
| 19 | seed area | 27 | 4 | 98 | 73 | 0 | 0 | 75 | 78 | 2.9 |
| 20 | seed area | 27 | 4 | 99 | 75 | 0 | 0 | 75 | 143 | 5.3 |
| 21 | seed area | 27 | 4 | 98 | 74 | 0 | 0 | 75 | 335 | 11.5 |
| 22 | seed area | 27 | 4 | 102 | 69 | 0 | 0 | 75 | 714 | 26.8 |
| 23 | seed area | 27 | 4 | 97 | 68 | 0 | 0 | 75 | 1460 | 51.1 |
| 24 | seed area | 42 | 4 | 99 | 71 | 0 | 0 | 75 | 2920 | 101 |
| 25 | seed area | 27 | 4 | 101 | 82 | 0 | 0 | 120 | 7400 | 304 |
| 26 | filter | 165 | 0 | 0 | 0 | 0 | 145 | 0 | 0 | 0 |
| 27 | filter | 100 | 10 | 520 | 175 | 0 | 0 | 60 | 0 | 0 |

Initial conditions for all chamber experiments are described in **Table 1**. The chamber was cleaned prior to each experiment by continually flushing with custom-filtered zero air, quantified to be below the detection limits of all available analyzers, for at least 12 hours (>7 full air exchanges). Two types of experiments were conducted: those using $O_3 + NO_2$ as an NO_3 source, and those using $N_2O_5$ as an $NO_3$ source. For the former, variable concentrations of $NO_2$ (diluted from 2042 ppm ±1% in $N_2$, Praxair) and $O_3$ were injected to initiate the formation of low levels of $NO_3$, according to the desired $nRO_2$ fate. $H_2O_2$ (50 wt.% in $H_2O$, Aldrich) was then injected by flowing 4 L min$^{-1}$ of ultra zero air through a bubbler warmed to 40 °C in a water bath. After the inorganic gas-phase reactants were introduced, the chamber was allowed to mix for 1-2 h, during which time secondary formation of $NO_3$ (from $O_3 + NO_2$) and $HO_2$ (from $NO_3 + H_2O_2$) could proceed. Seed particles were introduced to the chamber during mixing by atomizing a solution of 0.06 M ammonium sulfate (($NH_4)_2SO_4$, ≥99%, Aldrich) through a $^{210}$Po neutralizer. Lastly, liquid standards of α-pinene (Sigma Aldrich, >99%) were injected with gas-tight syringes into an airtight glass bulb, and introduced to the chamber by a 4 L min$^{-1}$ flow of ultra zero air. The α-pinene reacts quickly and thus was mixed rapidly from pulsed injections of high pressure ultra zero air (100 psi) for 2 min in order to initiate the reaction. The ozonolysis of α-pinene occurs concurrently with its $NO_3$-initiated oxidation with this experiment design. Ozonolysis of α-pinene produces OH, which reacts with $H_2O_2$ to be an additional dark formation source of $HO_2$, as well as with α-pinene itself. Control reactions were performed to accompany each experiment listed in **Table 1** that includes ozone, using the same conditions minus $NO_2$, in order to subtract out the SOA and other product formation from the purely ozonolytic reaction.

For experiments that used $N_2O_5$ as the $NO_3$ source, injections of α-pinene and other desired inorganic reactants, including seed particles, $NO_2$, and NO (200 ppm ±1% in $N_2$, Praxair), were conducted first. The reactions were then initiated by the rapid injection of gas-phase $N_2O_5$, which was previously evaporated into an evacuated 500 mL glass bulb to the desired pressure and backfilled to room pressure with $N_2$. $N_2O_5$ was synthesized according to Claflin and Ziemann (2018), verified using FT-IR, and stored in the dark at -20°C prior to use. In these experiments, the decomposition equilibria of $N_2O_5 \leftrightarrows NO_2 + NO_3$ was manipulated via injections of $NO_2$ in order to slow $NO_3$ formation and thus control the $nRO_2$ fate.

During experiments, mixing ratios of oxygenated gas-phase organics were quantified with a custom-built triple-quadrupole chemical ionization mass spectrometer (CIMS) using $CF_3O^-$ as the reagent ion. Instrumental details, including humidity-dependent calibration methods, have been described in detail previously (Crounse et al., 2006; St. Clair et al., 2010; Nguyen et al., 2014b; Praske et al., 2015). The CIMS detects PNP and other polar analytes predominantly without fragmentation as clusters with $CF_3O^-$. Although authentic standards of PNP are not available for direct calibration in CIMS, the analytical sensitivity of synthesized organic nitrates of different carbon length and neighboring groups in the $CF_3O^-$ CIMS were found to be different from each other by a factor of 20-30% (Lee et al., 2014; Teng et al., 2017); thus, sensitivity of PNP and pinene hydroxy-nitrate were assumed to be the same as isoprene hydroxynitrates with 30% uncertainty.

A scanning mobility particle sizer (SMPS), comprised of an electrostatic classifier (TSI 3080) and a condensation particle counter (TSI 3772), was used to measure particle size distributions between 15 nm and 670 nm. Control experiments monitoring the dark decay of ammonium sulfate seed aerosol concentrations in the chamber were used to determine diameter-

dependent particle wall loss rates (Schwantes et al., 2019), which were then used to correct experiment particle concentrations. Calculations of particle mass from measured aerodynamic diameter assume a density of 1.2 g cm$^{-3}$.

## 2.2 Kinetic modelling

We use a kinetic model (**Mech. S1**) to simulate gas-phase chemistry for each experiment in the environmental chamber. The mechanism uses reaction parameters from the JPL Chemical Kinetics and Photochemical Data Evaluation
(Burkholder et al., 2019), and is run on Matlab (MathWorks, Inc.). We also include reactions of α-pinene with OH, O$_3$, and NO$_3$, and isomer-specific reactions of the subsequently produced peroxy radicals with HO$_2$, NO, NO$_3$, and other RO$_2$ radicals (individually represented). Product yields from each pathway, along with rates of RO$_2$ + RO$_2$ reactions, are adjusted to fit the experimental data. We initialize simulations with the inorganic species listed in **Table 1** and allow the model to run for the allotted mixing time before instantaneously adding α-pinene. The model does not include any wall deposition of vapors or
gas-particle interactions, and is used only to estimate the concentrations of gas-phase species and the contributions of each peroxy radical reactive pathway.

## 2.3 SOA composition analysis by high-resolution mass spectrometry (HRMS)

SOA were collected for composition analysis using Omnipore hydrophilic Teflon filters (0.2 μm diameter pore, Millipore Corp.) that is compatible with polar and non-polar organics. The filters were gently extracted using LC-MS grade
acetonitrile (Optima, Fisher Scientific) to mass concentrations on the order of 100 μg/mL, depending on the experiment, by several ultrasound pulses of duration 1 s in order to limit cavitation in the ultrasonic bath that may alter analyte compositions. Filter extracts were directly infused into a linear-trap-quadrupole (LTQ) Orbitrap XL mass spectrometer (Thermo Instrument Corp.) using positive and negative ion mode electrospray ionization (ESI) at 4 kV spray voltage and a mass resolving power of 60,000 m/Δm at m/z 400. An external calibration was performed in both ion modes using commercial mass standards in the
range of 100 – 2000 m/z (Pierce™ LTQ ESI Positive and Pierce™ LTQ ESI Negative calibration solutions, Fisher Scientific), and the data were recalibrated until the mass accuracy obtained from standard solutions was < 1 ppm.

The data analysis was performed similarly to our previous works (Nguyen et al., 2010; Nguyen et al., 2011). Briefly, the raw data were de-convoluted using Decon2LS (freeware from Pacific Northwest National Laboratory), and background subtracted for peaks present in the solvent. The *m/z* peaks were assigned to molecular formulas (C$_c$H$_h$O$_o$N$_n$) based on a custom
Matlab script that applies Lewis and Senior rules (Kind and Fiehn, 2007) and a Kendrick Mass Defect analysis (base CH$_2$; Roach et al., 2011) that have been demonstrated on SOA mixtures. The prevalent ionization mechanism for this specific analyte mixture was found to be sodium cluster formation (M+Na$^+$) in the positive mode, which occurs preferentially for carbonyls (Kruve et al., 2013), and nitrate cluster formation (M+NO$_3^-$) in the negative mode, which is efficient for organic nitrates, alcohols, and other functional groups (Sisco and Forbes, 2015; Mathis and McCord, 2005). The nitrate anion, prominently
detected in the mass analyzer at m/z 61.988, was not purposefully introduced but likely formed either from in-source

fragmentation of organic nitrates or from $HNO_3$ produced during the hydrolysis of tertiary organonitrates from the aqueous LC solvents, and fortuitously acted as a reagent ion for chemical-assisted electrospray. While peak heights correlate well with concentration in direct-infusion ESI HRMS when the analyte matrix is similar (Chan et al., 2020), the correlation coefficients are unknown for each analyte in the mixture; thus, the HRMS data is qualitative. Furthermore, due to the labile $-ONO_2$ groups,
the organonitrate observations from HRMS likely represent a lower limit.

## 3 Results and discussion

### 3.1 SOA and PNP yields from different $n$RO$_2$ fates

To investigate the dependence of aerosol and gaseous PNP yields on the $n$RO$_2$ reaction partner, we performed a series of environmental chamber experiments (experiments 1-16, **Table 1**) with various starting conditions designed to isolate or
maximize the contributions of each RO$_2$ reaction pathway. Example time profiles from three representative experiments are shown in **Figure 1**. $n$RO$_2$ + NO$_3$, $n$RO$_2$ + NO, and $n$RO$_2$ + RO$_2$ chemistry were isolated in experiments using $N_2O_5$ as the NO$_3$

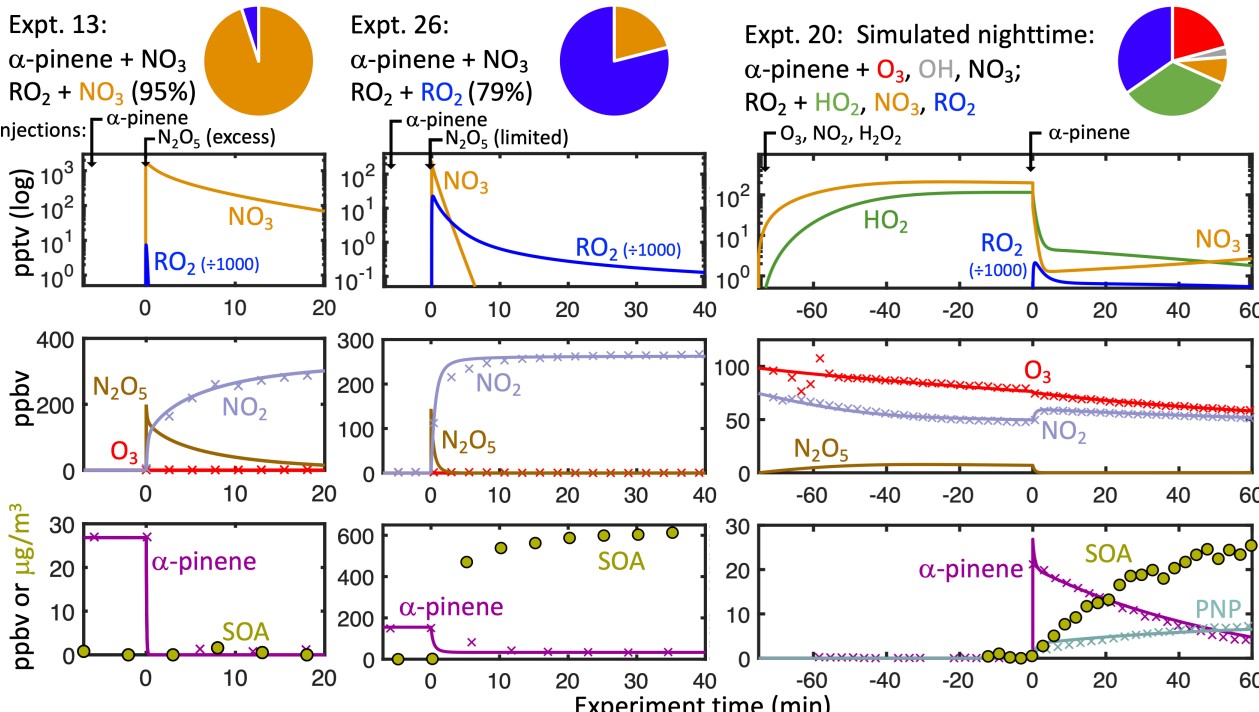

**Figure 1.** Modelled (solid lines) and measured (points) concentrations of key reactive species and products in three example chamber experiments. Timing of chamber injections are demarcated at top. Experiment time of 0 corresponds to the beginning of a-pinene oxidation. Pie charts show the percent contribution of each α-pinene oxidative pathway (α-pinene + O$_3$ in red, α-pinene + OH in grey and α-pinene + NO$_3$ speciated by subsequent RO$_2$ reaction partner: orange for NO$_3$, blue for RO$_2$, green for HO$_2$).

source. Using excess $N_2O_5$ as a source causes a rapid initial spike of $NO_3$, making the $nRO_2$ + $NO_3$ pathway dominant in Experiment 13 (**Fig. 1**, left). The $nRO_2$ + NO pathway is similarly easy to isolate with excess NO added before $N_2O_5$ injection, as in Experiment 15 (**Fig. S1**). $nRO_2$ + $nRO_2$ chemistry can be maximized by injecting excess α-pinene prior to $N_2O_5$ addition; the $NO_3$ is thus dominantly consumed by reaction with α-pinene, leaving the subsequently produced $nRO_2$ to react with each other, as in Experiment 26 (**Fig. 1**, middle), for which we calculate that 79% of $nRO_2$ reacted with other $nRO_2$.

$nRO_2$ + $HO_2$ chemistry is more difficult to isolate due to the scarcity of clean $HO_x$ sources in the dark. Here, we describe a method whereby reaction with $HO_2$ represents a majority of the $nRO_2$ fate without additional carbon inputs, initiated by the slow production of $NO_3$ in situ via $NO_2$ + $O_3$, which will introduce three reaction partners for α-pinene ($NO_3$, $O_3$, OH). Alkene ozonolysis has been used previously to produce dark OH in chamber experiments (Xu et al., 2019); Leveraging the ozonolysis of α-pinene already occurring in the chamber, we can amplify $HO_2$ production by injecting excess $H_2O_2$ prior to α-pinene to scavenge OH. Thus, the OH + $H_2O_2$ → $HO_2$ + $H_2O$ reaction simultaneously produces $HO_2$ and suppresses the side chemistry of α-pinene + OH. This $HO_2$ source is relevant to the nighttime atmosphere, as ozonolysis always occurs with $NO_3$ reaction in the ambient due to the major source chemistry from $NO_2$ + $O_3$.

The added $H_2O_2$ provides an additional benefit – its reaction with $NO_3$ is a source of $HO_2$ prior to α-pinene injection (**Fig. S2**). The $H_2O_2$ + $NO_3$ → $HO_2$ + $HNO_3$ reaction has been estimated by Burrows, Tyndall, and Moortgat (1985) to have an upper limit of $<2\times10^{-15}$ cm$^3$ molec$^{-1}$ s$^{-1}$ – too slow for atmospheric relevance, but sufficient to produce significant $HO_2$ in chambers when $H_2O_2$ is in excess. The rate coefficient of this reaction was further constrained from its upper limit based on the ratio of $k_{CH2O+OH}/k_{CH2O+NO3}$ (Burkholder et al., 2019), resulting in a rate coefficient of $1.1\times10^{-16}$ cm$^3$ molec$^{-1}$ s$^{-1}$ used in our simulations. The formation of PNP was not adequately reproduced if $HO_2$ is assume to originate from $H_2O_2$ + OH alone, i.e., omitting the $H_2O_2$ + $NO_3$ reaction in the kinetic model; the yield of PNP would need to be unphysically high to reconcile the difference.

The α-pinene + $NO_3$ reaction is modeled to be the major α-pinene sink in experiments initiated by $NO_2$ + $O_3$ (e.g., 73% in Experiment 22, **Fig. 2**, **Table 2**). Even so, the background chemistry from α-pinene + $O_3$ and α-pinene + OH (from the ozone control experiments without $NO_2$) are substantial sources of SOA and require careful subtraction (**Fig. 2**). Caveats to this approach include: (1) in the controls, ozone and OH are larger sinks for α-pinene due to a lack of competition from $NO_3$; thus, a larger fraction

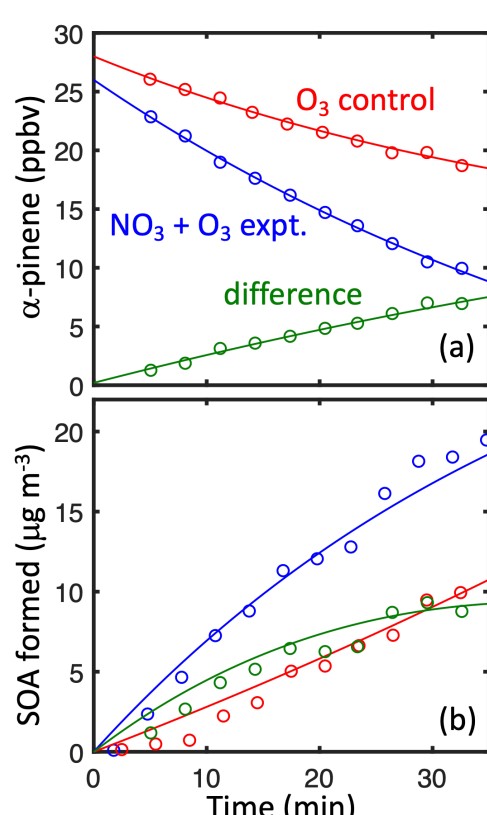

**Figure 2.** (a) α-pinene loss and (b) SOA formation in 'simulated nighttime' Experiment 22 (blue) and the ozonolysis control experiment (red). The difference (green) shows the contribution from α-pinene + $NO_3$.

of α-pinene produces SOA from ozonolysis in the control compared to the experiment and the subsequent subtraction obtains a lower-limit SOA yield; (2) synergistic reactions between $RO_2$ intermediates from the different oxidation pathways are not possible to isolate, and contributed roughly 20% to the analytical signal from the SOA composition analysis (**Section 3.3**); however, it is now appreciated that these synergies also occur in the ambient and are not realistic to ignore in laboratory and modeling studies (Kenseth et al., 2018; Inomata, 2021).

These 'simulated nighttime' experiments (e.g. Experiment 20, **Fig. 1**, right) are termed as such because they provide an atmospherically relevant balance of reactive pathways, in which fractional contributions are comparable to those on summer nights in the Southeast United States (60–80% α-pinene + $NO_3$, 20–40% α-pinene + $O_3$; 30–50% $nRO_2$ + $HO_2$, 30–50% $nRO_2$ + $RO_2$; Ayres et al., 2015; Romer et al., 2018). This is corroborated by gas phase data from the CIMS (**Fig. 3**), which show that $NO_3$-initiated products are the major

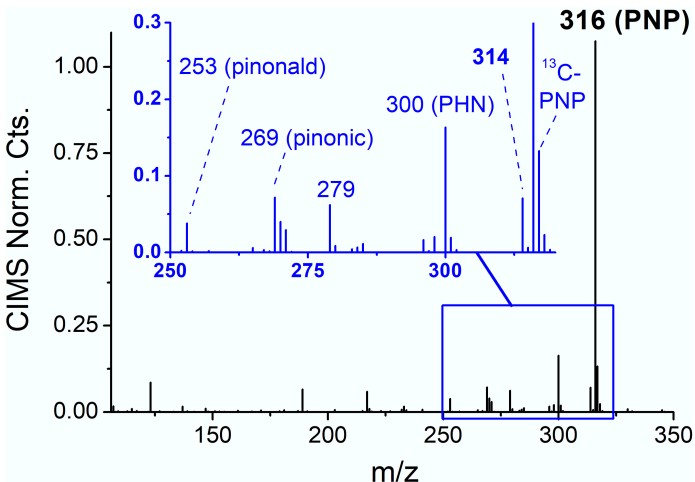

**Figure 3.** Negative ion CIMS mass spectrum after 1 h of oxidation under 'simulated nighttime' conditions, showing PNP, PHN, and another organic nitrogen product at $m/z$ 314. Pinonaldehyde is observed at $m/z$ 253 and pinonic acid at $m/z$ 269.

volatile products formed in the simulated nighttime experiments. Furthermore, these experiments are able to reproduce gaseous organonitrates observed during the Southern Oxidant and Aerosol Study (SOAS; Carlton et al., 2018), including PNP – a major monoterpene semivolatile at the SOAS site in rural Alabama – and an unknown compound at $m/z$ 314 that has also been observed in the ambient with similar time profiles to PNP (**Fig. S3**).

Product yield results from different $nRO_2$ reaction pathways are shown in **Table 2**, along with the modeled contributions of each $nRO_2$ reaction pathway. We observe low SOA formation in experiments maximizing $nRO_2$ + $NO_3$ (3(±3)% mass yield, Experiment 13) and $nRO_2$ + NO (12(±7)%, Experiment 15). This is consistent with previous observations of 0–16% SOA mass yields from $N_2O_5$-initiated experiments (in which the $nRO_2$ + $NO_3$ pathway tends to dominate), where the larger yields were obtained only with slow continuous introductions of $N_2O_5$ (Bell et al., 2021; Fry et al., 2014; Hallquist et al., 1999; Moldanova et al., 2000; Muutzel et al., 2021; Spittler et al., 2006). We also observe no correlation between SOA yields in 'simulated nighttime' experiments and the fractional contribution of the $nRO_2$ + $HO_2$ pathway (**Fig. S4**), suggesting no SOA production from this pathway, consistent with Kurten et al. (2017) and Nah et al. (2016). This result conforms to expectations as $nRO_2$ + $HO_2$ produces either the RO radical (Iyer et al., 2018) leading to the volatile pinonaldehyde (Kurtén et al., 2017) or the hydroperoxide, which has high enough volatility to be observed by the CIMS. We do, however, observe high yields of SOA in experiments targeting $nRO_2$ + $nRO_2$ chemistry (up to 82% mass yield, Experiment 26). This includes both $N_2O_5$-initiated experiments with excess α-pinene and $O_3$ + $NO_2$ + $H_2O_2$-initiated 'simulated nighttime' experiments, during which we measured SOA mass yields of 19–55%.

**Table 2.** Peroxy radical pathway contributions and measured experimental outcomes for experiments performed in this work.

| Expt | $X(HO_2)_i$ (ppt) | $X(NO_3)_i$ (ppt) | $X(RO_2)_{max}$ (ppb) | pinene +NO$_3$ (%) | $n$RO$_2$ +NO$_3$ (%) | $n$RO$_2$ +NO (%) | $n$RO$_2$ +HO$_2$ (%) | $n$RO$_2$ +RO$_2$ (%) | SOA mass yield (%)[a] | PNP molar yield (%) |
|---|---|---|---|---|---|---|---|---|---|---|
| | | | | Modelled | | | | | Measured | |
| 1 | 0 | 259 | 2.9 | 60 | 11 | 0 | 0 | 89 | 26.9 | 0 |
| 2 | 0 | 542 | 4.7 | 85 | 31 | 0 | 0 | 69 | 28.9 | 0 |
| 3 | 73 | 160 | 3.1 | 95 | 19 | 0 | 37 | 44 | 18.3 | 17 |
| 4 | 121 | 114 | 1.9 | 89 | 10 | 0 | 65 | 25 | 9.2 | 31 |
| 5 | 120 | 303 | 4.0 | 99 | 37 | 0 | 41 | 21 | 2.5 | 19 |
| 6 | 97 | 56 | 0.94 | 74 | 5 | 0 | 70 | 25 | 5.1 | 30 |
| 7 | 34 | 152 | 3.3 | 90 | 17 | 0 | 18 | 65 | 22 | 9 |
| 8 | 147 | 151 | 2.0 | 89 | 11 | 0 | 66 | 23 | 10.5 | 32 |
| 9 | 37 | 159 | 4.0 | 69 | 6 | 0 | 41 | 53 | 22.1 | 17 |
| 10 | 35 | 151 | 4.6 | 46 | 2 | 0 | 22 | 76 | 23.2 | 9 |
| 11 | 88 | 277 | 3.1 | 81 | 16 | 0 | 23 | 61 | 26.4 | 10 |
| 12 | 0 | 0 | 6.0 | 100 | 41 | 0 | 0 | 59 | 67 | 0 |
| 13 | 0 | 0 | 7.4 | 100 | 95 | 0 | 0 | 5 | 3 | 0 |
| 14 | 0 | 0 | 5.6 | 100 | 8 | 53 | 0 | 39 | 43 | 0 |
| 15 | 0 | 0 | 0.01 | 100 | 0 | 100 | 0 | 0 | 12 | 0 |
| 16 | 0 | 0 | 3.0 | 100 | 7 | 59 | 0 | 33 | 33 | 0 |
| 17 | 113 | 194 | 2.1 | 75 | 10 | 0 | 45 | 45 | 0 | 20 |
| 18 | 111 | 186 | 2.0 | 75 | 10 | 0 | 45 | 45 | 4 | 21 |
| 19 | 112 | 191 | 2.1 | 75 | 10 | 0 | 44 | 45 | 5.7 | 21 |
| 20 | 114 | 195 | 2.1 | 76 | 11 | 0 | 44 | 45 | 21 | 26 |
| 21 | 113 | 192 | 2.1 | 76 | 10 | 0 | 45 | 45 | 21.9 | 25 |
| 22 | 112 | 188 | 2.0 | 73 | 10 | 0 | 45 | 45 | 19.9 | 25 |
| 23 | 110 | 181 | 1.9 | 71 | 9 | 0 | 45 | 45 | 19 | 21 |
| 24 | 112 | 189 | 2.3 | 60 | 6 | 0 | 46 | 48 | 21.6 | 20 |
| 25 | 110 | 178 | 2.0 | 75 | 10 | 0 | 44 | 46 | 23.5 | 18 |
| 26 | 0 | 0 | 23 | 100 | 21 | 0 | 0 | 79 | 82 | 0 |
| 27 | 295 | 504 | 5.5 | 64 | 12 | 0 | 41 | 47 | 55 | [b] |

[a]ozonolysis-corrected (see Table S1); [b]not measured

Across all experiments, the modeled fractional contribution of the $n$RO$_2$ + RO$_2$ pathway correlates strongly with SOA yield (**Fig. 4**). An error-weighted linear regression (York et al., 2004) of ozonolysis-corrected SOA yield against the modeled $n$RO$_2$ + RO$_2$ contribution for seeded experiments suggests a 58($\pm$6)% mass yield ($R^2$ = 0.54) from this pathway alone. With such a large number of experiments spanning a range of different pathway contributions, however, we can improve upon this

simple fit by performing a multivariate linear regression to estimate SOA yields from all pathways, with coefficients limited to be $\geq 0$. This results in a predicted SOA mass yield from the $n$RO$_2$ + RO$_2$ pathway identical to results from the simple linear
estimate within error: 56($\pm$7)%. SOA mass yields from the other pathways are not significantly different from zero: 13($\pm$11)% for $n$RO$_2$ + NO, 4($\pm$8)% from $n$RO$_2$ + NO$_3$, 0($\pm$5)% from $n$RO$_2$ + HO$_2$. Results from this regression analysis, which gives an R$^2$ of 0.66 against measured SOA yields, are shown in **Figure 4**.
Repeating the regression analysis for all $\alpha$-pinene sinks in the 'simulated nighttime' experiments, including the ozonolysis and OH pathways as additional independent variables, also gives coefficients indistinguishable within error (0% for $\alpha$-pinene + OH, 15% for $\alpha$-pinene + O$_3$).

By inherently treating individual reactive pathways as independent variables, these regression analyses cannot separate

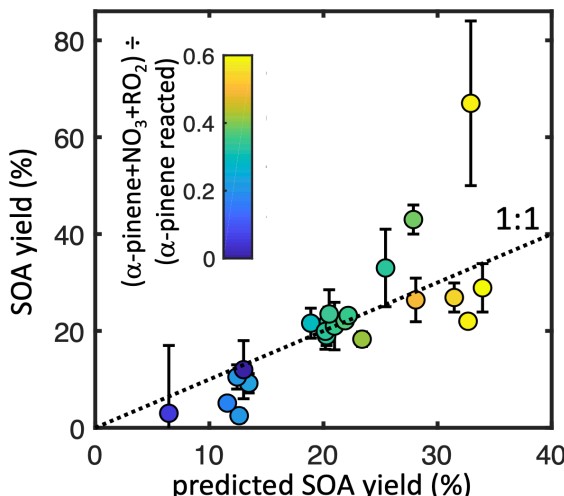

**Figure 4.** Measured (non-ozonolysis-corrected) SOA mass yields from each experiment as a function of the modelled contribution of $n$RO$_2$ + $n$RO$_2$ (color) and of the predicted SOA yield from the sum of each pathway contribution (x axis). Error bars denote uncertainty from SMPS measurements.

the possible contributions of interactions between multiple pathways, *e.g.* from synergistic dimerization between $n$RO$_2$ and ozonolysis-derived RO$_2$ (see **Section 3.3**). The reported coefficients may therefore misrepresent what each pathway on its own would contribute to SOA formation without such synergy. However, because the analysis was performed on experiments
predominantly designed to replicate the reactive pathways experienced by a-pinene under ambient nighttime conditions, we expect the coefficients reported above to represent reasonable estimates of each pathway's contribution to SOA formation in the nighttime atmosphere, including any synergistic reactive pathways that occur in ambient conditions. Conditions which deviate substantially from the nighttime atmosphere and therefore lack the same synergistic reactions (*e.g.* experiments isolating single reactive pathways) may measure different SOA yields. Among the experiments performed here, those with
high $n$RO$_2$ + RO$_2$ contributions without any ozonolysis (Experiments 12, 14, and 26) exhibited some of the highest measured SOA yields – higher than the regression model would predict – suggesting perhaps that the $n$RO$_2$ + $n$RO$_2$ pathway on its own results in even higher SOA yields while $n$RO$_2$ + other RO$_2$ pathways have lower yields. Without knowing the relative rates of various $n$RO$_2$ + RO$_2$ reactions, though, we cannot sufficiently constrain these differences, and additional regression analyses including interaction terms between the reactive pathways did not yield statistically robust results.
These regression analyses also intrinsically depend on kinetic model parameters such as bimolecular RO$_2$ reaction rates, some of which are uncertain. While we are unable to fully quantify these rates, we find that certain ratios between rates are constrained by our experimental outcomes. For example, the negligible SOA yield in the high-NO$_3$ Experiment 13 suggests that $n$RO$_2$ + RO$_2$ chemistry cannot play a major role in that experiment. This can only be achieved if the bulk rate coefficient for $n$RO$_2$ + RO$_2$ reactions ($k_{\text{RO2+RO2}}$) is approximately two orders of magnitude slower than that of $n$RO$_2$ + NO$_3$ reactions

($k_{RO2+NO3}$). We therefore set the bulk $k_{RO2+RO2}$ to $1\times10^{-13}$ cm$^3$ molecule$^{-1}$ s$^{-1}$ with the corresponding bulk $k_{RO2+NO3}$ equal to $1\times10^{-11}$ cm$^3$ molecule$^{-1}$ s$^{-1}$ (a value higher than most reported $k_{RO2+NO3}$, although few measurements exist for these reactions). The PNP yields in 'simulated nighttime' experiments also provide a constraint on the relative rates of the nRO$_2$ + RO$_2$ and nRO$_2$ + HO$_2$ reactions. RO$_2$ + HO$_2$ rate coefficients are, in general, are better constrained than those of RO$_2$ + RO$_2$ or RO$_2$ + NO$_3$ reactions; we use the parameterization in Wennberg et al. (2018), which gives $1.7\times10^{-11}$ cm$^3$ molecule$^{-1}$ s$^{-1}$, similar to the

measured value for pinene-derived radicals of $2.1\times10^{-11}$ cm$^3$ molecule$^{-1}$ s$^{-1}$ from Boyd et al. (2003). Increasing $k_{RO2+RO2}$ in our simulations causes a decrease in the contribution of RO$_2$ + HO$_2$ chemistry, but this decrease is only plausible until the measured PNP yield is 100%, which occurs at a $k_{RO2+RO2}$ value slightly above $1\times10^{-12}$ cm$^3$ molecule$^{-1}$ s$^{-1}$. The

substantial yield of pinonaldehyde instead of PNP from nRO$_2$ + HO$_2$ calculated by Kurtén et al. (2017) suggests that $k_{RO2+RO2}$ must be well below $1\times10^{-12}$ cm$^3$ molecule$^{-1}$ s$^{-1}$ to keep the PNP yield well below 100%.

       Based on these constraints, a value of $1\times10^{-13}$ cm$^3$ molecule$^{-1}$

s$^{-1}$ represents our optimal fit and a value of $1\times10^{-12}$ cm$^3$ molecule$^{-1}$ s$^{-1}$ represents an upper limit for the bulk $k_{RO2+RO2}$ of α-pinene nRO$_2$. A higher value would require implausibly high $k_{RO2+NO3}$ values in order to maintain dominance of the $n$RO$_2$ + NO$_3$ pathway in Experiment 13, and would require implausibly high $k_{RO2+HO2}$ values in order to maintain a

measured PNP branching ratio below the maximum of 100%. Our best estimate is reasonably within the broad range of measured $k_{RO2+RO2}$ for other peroxy radicals, particularly given that rate coefficients tend to decrease with size and degree of substitution at the radical site (Orlando and Tyndall, 2012). It is smaller, however, than some recent

measurements of the $k_{RO2+RO2}$ of peroxy radicals derived from α-pinene ozonolysis ($1\times10^{-12}$ to $1\times10^{-11}$ cm$^3$ molecule$^{-1}$ s$^{-1}$), highlighting the uncertainty in reaction rates of this type (Zhao et al., 2018; Berndt et al., 2018b). A slower bulk $k_{RO2+RO2}$ of $1\times10^{-14}$ cm$^3$ molecule$^{-1}$ s$^{-1}$, which would still be consistent with our constraints on relative rate ratios,

would scale the contribution of nRO$_2$ + RO$_2$ chemistry in the 'simulated nighttime' experiments by a factor of 0.67, which in turn would require scaling SOA yields from this pathway up by 50%. A faster bulk $k_{RO2+RO2}$ of $1\times10^{-12}$ cm$^3$ molecule$^{-1}$ s$^{-1}$ would scale the $n$RO$_2$ + RO$_2$ contribution

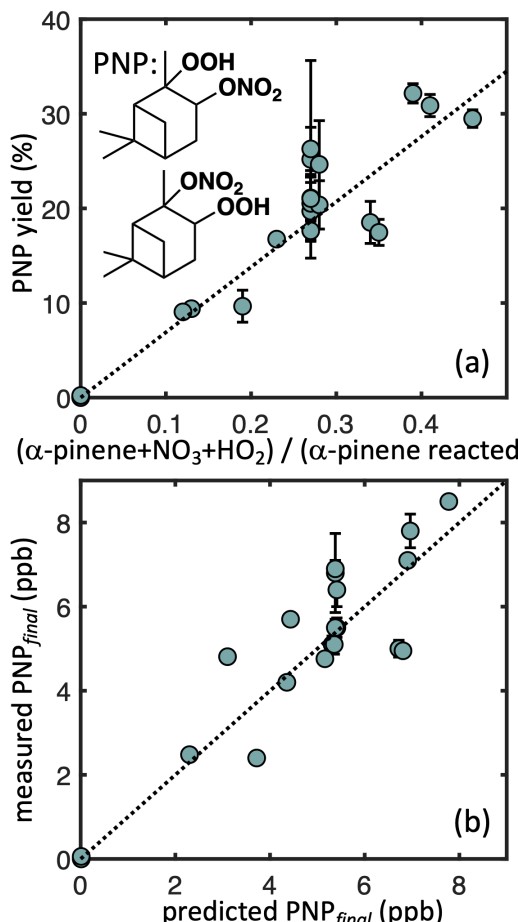

**Figure 5.** Measured α-pinene nitrooxy-hydroperoxide (PNP) yields as a function of the modelled contribution of the $n$RO$_2$ + HO$_2$ pathway (top) and measured PNP at the end of each experiment as a function of the predicted PNP based on the simple kinetic model. Dotted lines denote error-weighted ordinary least squares fits with intercepts constrained to zero (R$^2$ = 0.88 top, 0.74 bottom; slope = 0.58 top, 1.0 bottom). Error bars denote uncertainty from CIMS measurements.

in the same experiments by a factor of 1.16, which in turn would require scaling the $nRO_2 + RO_2$ SOA yield down by 14%.

Despite our constraints, the bulk $k_{RO2+RO2}$ remains highly uncertain, and is also likely to differ between $nRO_2$ isomers for self- and cross-reactions (Orlando and Tyndall, 2012).

To constrain the sequestration of reactive nitrogen under ambient conditions from the α-pinene + $NO_3$ reaction, we also quantify PNP yields in our simulated nighttime experiments. **Figure 5** (top) shows measured PNP molar yields as a function of the $nRO_2 + HO_2$ fate contribution. Regression analysis (York et al., 2004) suggests a branching ratio for PNP

formation of 58(±2)% from $nRO_2 + HO_2$ ($R^2 = 0.88$), which we implement in our kinetic model (**Fig. 5**, bottom). Including the uncertainty in the CIMS calibration increases the uncertainty bounds to 58(±20)%. The bulk yield likely represents a combination of different branching ratios to PNP formation from the different $nRO_2$ isomers. We hypothesize that the secondary (minor) $nRO_2$ produces exclusively PNP in its reaction with $HO_2$, which would imply a branching ratio of 37% for PNP formation from the tertiary (major) $nRO_2 + HO_2$, assuming an initial branching of 65:35 major:minor isomers from α-

pinene + $NO_3$ (Jenkin et al., 1997; Saunders et al., 2003). Our measured bulk PNP yield is somewhat higher than the 30% estimated in the FIXCIT studies (Kurtén et al., 2017), which used similar instrumentation but slightly different chamber conditions (formaldehyde instead of $H_2O_2$ and slow addition of α-pinene to minimize $RO_2 + RO_2$ chemistry). Some of this discrepancy might be explained by an overestimate of the contribution of $nRO_2 + HO_2$ chemistry in the FIXCIT experiment, where they assumed that all $NO_3$ produced by $O_3 + NO_2$ reacted with α-pinene and that all $nRO_2$ reacted with $HO_2$ (Kurtén et

al., 2017).

We also detect α-pinene hydroxy nitrate (PHN), a product of $nRO_2 + nRO_2$ chemistry, and α-pinene dinitrate (PDN), a product of $nRO_2 + NO$ or $nRO_2 + NO_3$ chemistry, by $CF_3O^-$ CIMS. Based on similar regression analyses of observations and modeling results (**Fig. S5**), we estimate a bulk PHN yield from $nRO_2 + nRO_2$ of 11.7(±3.3)%. However, because the major $nRO_2$ isomer (with the tertiary peroxy radical) is unable to donate an α-hydrogen for PHN formation, a bulk yield of 11.8%

represents a 34% branching ratio from the minor (secondary) $nRO_2$ isomer (assuming an initial ratio of 65:35 major:minor isomers from α-pinene + $NO_3$; Jenkin et al., 1997; Saunders et al., 2003). While observed PDN correlates with $nRO_2 + NO_3$ chemistry, its low signal and uncertain CIMS sensitivity makes quantification difficult.

### 3.2 SOA yield dependence on seed surface area

SOA from gas–particle partitioning tends to exhibit a strong dependence on seed aerosol surface area, because of

competition between particle surfaces and chamber walls (Schwantes et al., 2019; Zhang et al., 2014; Zhang et al., 2015). This effect can cause an underestimation of SOA yields in atmospheric chambers if insufficient seed aerosol is used. To quantify this effect and ensure that $RO_2$ fate experiments (#1-16) were initiated with sufficient seed aerosol, we performed an additional set of experiments (#17-25) under 'simulated nighttime' conditions to investigate the dependence of SOA yield on seed surface area (**Fig. 6**). No SOA formation was observed in the absence of seed aerosol, indicating that α-pinene + $NO_3$ does not nucleate.

This is in contrast to our ozonolysis control experiments where nucleation was observed, in agreement with other accounts

(Burkholder et al., 2007; Hoppel et al., 2001; Inomata, 2021; Takeuchi et al., 2019). The lack of nucleation is consistent with the expectation that the α-pinene + NO₃ reaction dominates the nighttime experiments over ozonolysis. SOA yields reached a maximum when initial seed surface area reached 143 $\mu m^2$ $cm^{-3}$ (equivalent under experimental conditions to 5.3 $\mu g$ $m^{-3}$). This threshold is at least an order of magnitude lower than those measured by Schwantes et al. (2019) for SOA from isoprene OH oxidation under high-NO conditions and by Zhang et al. (2014) for SOA from toluene oxidation. Although vapor wall losses may still cause a decrease in measured SOA yields regardless of seed surface area (Krechmer et al., 2020), we assume here that these effects are minimal at the high seed surface areas (> 600 $\mu m^2$ $cm^{-3}$) used in our experiments to investigate RO₂ fate. However, true SOA yields may be higher than reported if vapor wall losses are considered.

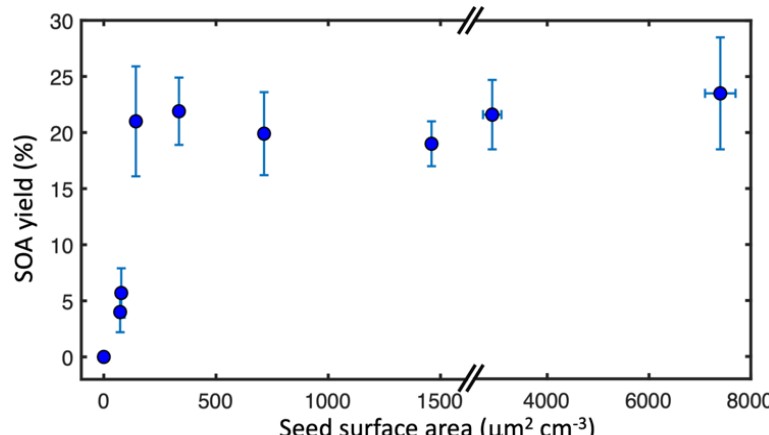

**Figure 6.** Measured ozonolysis-corrected SOA yields from seed area experiments (#17-25) plotted against initial seed surface area. Error bars denote experimental uncertainty from SMPS measurements.

### 3.3 SOA molecular composition

Filters collected following SOA formation in Experiments 26 (high $n$RO₂ + $n$RO₂), 27 (simulated nighttime), and the ozonolysis control (without NO₂) for Experiment 27 were analyzed by HRMS to determine the SOA molecular composition (**Figure 7**). As expected, the mass spectrum from the 'simulated nighttime' experiment (#27, **Fig. 7B**) exhibit substantial overlap with those from the N₂O₅-initiated experiment in which the $n$RO₂ + $n$RO₂ pathway dominated (#26, **Fig. 7D**), corroborating the results in **Section 3.1** that suggested the $n$RO₂ + $n$RO₂ pathway is responsible for most SOA formation from α-pinene + NO₃ chemistry. The SOA from $n$RO₂ + $n$RO₂ chemistry is dominated by compounds containing 17-20 carbon atoms, suggesting that dimerization is the primary mechanism by which $n$RO₂ + $n$RO₂ reactions lead to SOA formation.

Masses in the 'simulated nighttime' SOA that are also observed in the $n$RO₂ + $n$RO₂ control experiment (#26) account for 39% of the negative mode peak signal and 29% of the positive mode peak signal in the SOA from the 'simulated nighttime' experiment. Negative mode may overestimate the contributions from dinitrated dimers from $n$RO₂ + $n$RO₂ chemistry, whereas positive mode may underestimate them. In the positive mode, it was found that 34% of the overall peak signal in the 'simulated nighttime' experiment can also be observed from ozonolysis chemistry (including OH reactions), 14% of peak signal is found in both the ozonolysis and $n$RO₂ + $n$RO₂ control experiments, and 24% of overall peak signal is completely unique to the nighttime chemistry. SOA from α-pinene ozonolysis is dominated by oxygenated monomers in the negative mode and lacks organonitrate functionalities, making it easy to distinguish from the mass spectrum of NO₃-oxidized SOA. The peaks from α-

pinene ozonolysis can thus be subtracted off from the 'simulated nighttime' SOA mass spectrum, leaving a spectrum attributable to α-pinene + NO₃ under 'simulated nighttime' conditions (**Fig. 7C**).

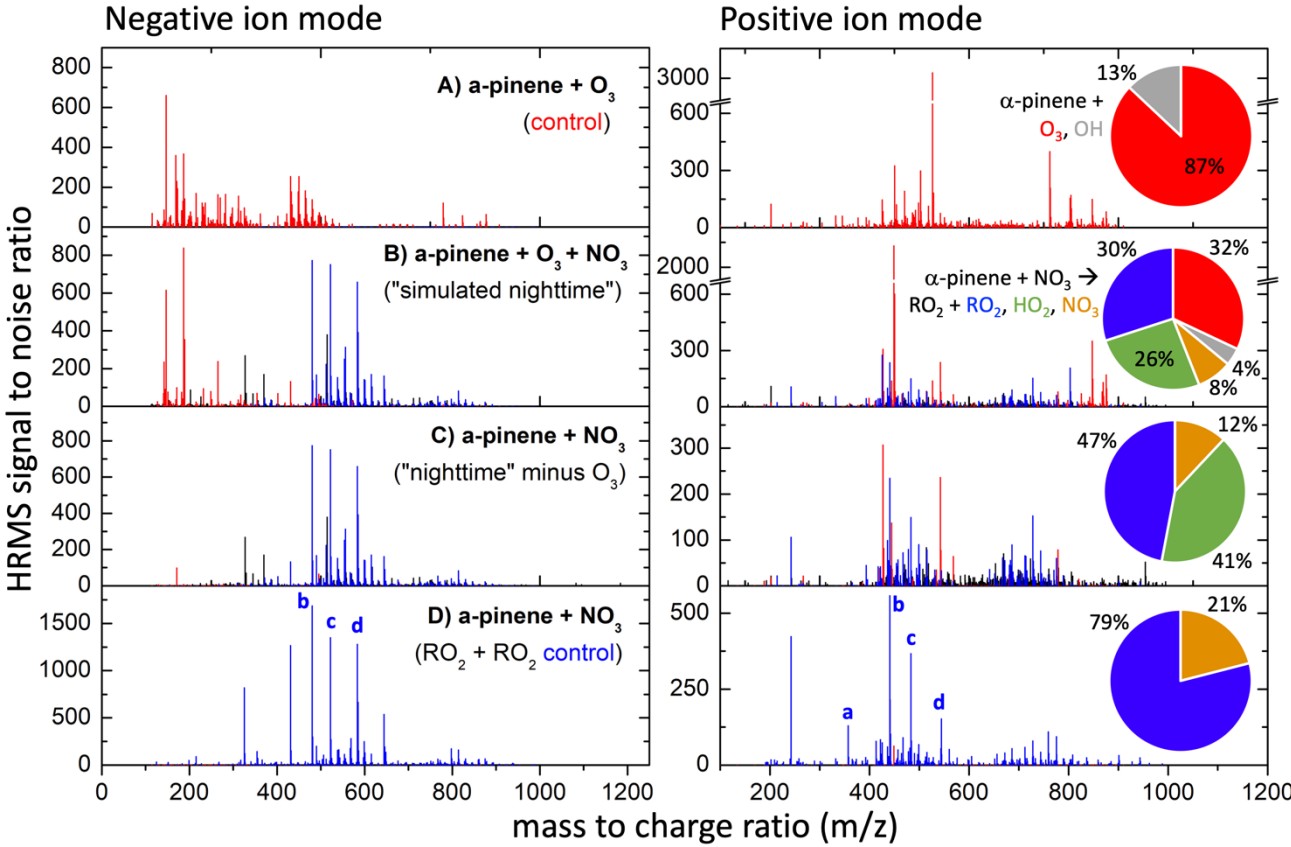


**Figure 7.** Mass spectra of filter samples from Experiments 26 (D) and 27 (B), along with the ozonolysis-only control for Experiment 27 (A) and the signal difference between B and A (C). Peaks uniquely attributable to α-pinene ozonolysis alone are shown in red, while those uniquely attributable to α-pinene + NO₃ followed by RO₂ + RO₂ chemistry are shown in blue, with all other peaks in black. Pie charts denote the fraction of α-pinene reacting via each pathway for a given experiment.
Formation mechanisms for the key dimer species (a-d) identified in the bottom panels are shown in Schemes 2-3.

The signal-weighted composition of SOA from the 'simulated nighttime' reaction (**Fig. 7B**), which includes background ozonolysis, is 60–70% nitrogen-containing organics with 1–3 nitrate groups, as determined in both negative and positive mode (where organic nitrogen species are not enhanced). Of the organic nitrogen, it appears that 2N species are the
most abundant and comprise ~30% of the SOA on average. This suggests dimerization from $nRO_2 + nRO_2$ to form $C_{14-20}$ compounds is highly important to SOA formation. Previous analyses of the molecular composition of SOA from α-pinene + NO₃ have also found a dominant contribution of dimers in the particle phase (Bell et al., 2021; Takeuchi et al., 2019). The 0N, 1N, and 3N species may also be dimers (4N were not highly observed), as each of the expected RO₂s from all channels can have 0–2 N (**Fig. 8**). There is also a clear population of trimers from α-pinene SOA that is observed here (**Fig. 7**, m/z centered

around 700, with associated carbons $C_{21}$-$C_{30}$) and elsewhere (Claflin and Ziemann, 2018; Romonosky et al., 2017). This may suggest $RO_2$s can react with neutral compounds to propagate $RO_2$ radicals that then terminate with another $RO_2$, or may alternatively be attributable to second-generation accretion reactions of dimers (especially those still containing a double bond, which would react rapidly with $NO_3$) with $RO_2$ monomers.

The data show that SOA formation from α-pinene + $NO_3$ originates from a varied cohort of distinct $nRO_2$ isomers.

In **Scheme 2** we suggest multiple pathways involving bimolecular reactions, β-scission, and intramolecular isomerization that can produce different $C_7$-$C_{10}$ $nRO_2$ isomers following the initial reaction of α-pinene with $NO_3$. These pathways are consistent with $RO_2$ chemistry previously suggested from theoretical (Kurtén et al., 2017) or lab studies (Xu et al., 2019) of similar systems. Some reactions of $nRO_2$ with NO, $NO_3$, or $HO_2$ shown in **Scheme 2** propagate $nRO_2$ radicals due to unimolecular decomposition rather than forming stable products; this implies that dimer-containing SOA from $nRO_2$ can originate from all

$nRO_2$ fate pathways. **Figure 8C** shows the most abundantly observed dimers in the SOA from $nRO_2$ + $nRO_2$ chemistry in Experiment 26 (labeled peaks a-d in **Fig. 7D**), which include $C_{20}H_{30}O_4$ (α-α, **Fig 7D.a**; presumably following the loss of 2 HONO from dimer $C_{20}H_{32}N_2O_8$ in the ESI source), $C_{17}H_{26}N_2O_{10}$ (δ-α, **Fig. 7D.b**), $C_{20}H_{32}N_2O_{10}$ (β-β, **Fig. 7D.c**), and $C_{20}H_{31}N_3O_{13}$ (ε-α, **Fig 7D.d**). Bell et al. (2021) also observed $C_{20}H_{32}N_2O_8$ and $C_{20}H_{32}N_2O_{10}$ among the most abundant compounds in SOA from α-pinene + $NO_3$ chemistry in which $nRO_2$ + $nRO_2$ reactions were thought to dominate.

**Scheme 2.** Proposed mechanisms for the formation of stable products (red) from α-pinene + $NO_3$ oxidation, including nitrooxy-hydroperoxide (PNP), α-pinene hydroxynitrate (PHN), pinene dinitrate (PDN), the oxidation product observed by CIMS at *m/z* 314, and pinonaldehyde, as well as α-pinene-derived peroxy radicals ($nRO_2$α-φ, blue) that may contribute to particle-phase dimers (**Fig. 8**) following the dominant secondary addition of $NO_3$ to α-pinene. "X" stands in for any of $RO_2$, $HO_2$, NO, or $NO_3$ as bimolecular reaction partners.

Dimers from the ozonolysis control experiments are notably absent from the 'simulated nighttime' experiment. This is likely because the peroxy radicals from α-pinene ozonolysis, which would have reacted with each other to dimerize in the ozonolysis-only experiment, instead react with the more abundant $nRO_2$ radicals from α-pinene + $NO_3$ in the 'simulated nighttime' experiment. This dimerization from cross-reactions of peroxy radicals from both oxidation pathways is responsible for the ~ 24% additional HRMS signals in the α-pinene + $NO_3$ SOA spectrum (**Fig. 7C**, black peaks) but not in the $nRO_2$ +

$nRO_2$ SOA spectrum (**Fig. 7D**). **Figure 8D** show how RO2s formed from ozonolysis (**i-iii**; Iyer et al., 2021), initially via the vinylhydroperoxide channel, and OH reactions (**1-2**; Xu et al., 2019) can couple with $nRO_2$s to produce synergistic dimers during nighttime oxidation of α-pinene. This type of synergy is missing when lab studies cleanly isolate reaction pathways, but may be an important part of SOA formation in the ambient as reactions occur simultaneously (Berndt et al., 2018a & b; Kenseth et al., 2018; Zhao et al., 2018; McFiggans et al., 2019; Inomata, 2021). These results underscore the importance of

conducting chamber experiments under atmospherically relevant conditions.

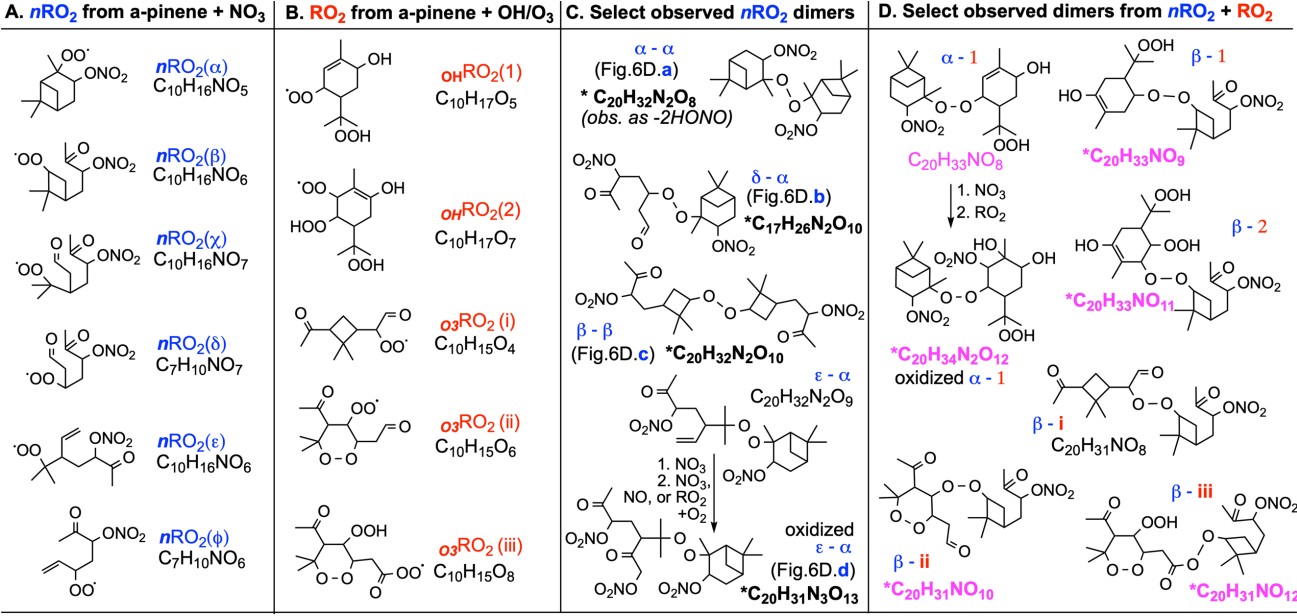

**Figure 8.** Structures and molecular formulas for (A) $nRO_2$ from $NO_3$-initiated chemistry (**Scheme 2**), (B) $RO_2$ from OH or $O_3$ chemistry (Xu et al., 2019; Iyer et al., 2021), (C) prominent dimers observed from $nRO_2$ self/cross reactions, and (D) dimers from synergistic $nRO_2$ + $RO_2$ couplings observed in the 'simulated nighttime; reactions. Dimers with chemical formulas written in pink are uniquely observed in the 'simulated nighttime' experiments; those marked with a * have moderate-to-high signal (40 > S/N).

       While we do not seek to represent each individual $nRO_2$ and dimerization pathway in our kinetic model, we adjust the product yields from the first-generation $nRO_2$ + $nRO_2$ reactions to provide a reasonable estimate of dimer formation. The observed combined positive and negative mode peak signal attributable to $C_{14-20}$ compounds (presumed dimers) in $nRO_2$ + $nRO_2$ SOA (Experiment 27, **Fig. 6D**) constitutes 40-60% of the total signal. To produce this 40-60% of the 56% SOA mass

yield estimated from $nRO_2$ + $nRO_2$ chemistry in **Section 3.1** using dimers, assuming a mean dimer molar mass of 485 g mol$^{-1}$

(the average of the positive mode and negative mode peak-signal-weighted mean), requires a molar yield of 0.16 dimer molecules from the $n\mathrm{RO_2} + n\mathrm{RO_2}$ reaction (0.09 per $n\mathrm{RO_2}$). This bulk effective yield represents an average across the various $n\mathrm{RO_2} + n\mathrm{RO_2}$ isomer permutations; branching ratios from individual $n\mathrm{RO_2} + n\mathrm{RO_2}$ isomer reactions may differ substantially, but in our kinetic model we simply apply the bulk effective yield to all peroxy radical self- and cross-reactions and scale down

the pinonaldehyde-forming pathway accordingly. This 16% dimer yield is somewhat higher than has previously been measured in other systems (Orlando and Tyndall, 2012), including from a-pinene ozonolysis (Zhao et al., 2018), but fits with recent evidence suggesting that increased molecular size and functionalization can significantly increase dimer branching ratios (Molteni et al., 2019; Berndt et al., 2018a). However, our estimated dimer yield is highly uncertain for a wide range of reasons, including (a) variations in compound-specific sensitivities within the ESI HRMS, which may accentuate the dimer contribution

as ESI efficiency has been shown to increase with molecular size (Kenseth et al., 2020); (b) sensitivity to the uncertain estimates of $\mathrm{RO_2}$ fate, which depend on poorly-constrained $\mathrm{RO_2} + \mathrm{RO_2}$ and $\mathrm{RO_2} + \mathrm{NO_3}$ rates (see **Section 3.1**); (c) the possibility of particle-phase reactions, or reactions during filter extraction, which may form or break dimers; and (d) lack of consideration of trimers, which account for 14% of overall peak intensity and may form from dimers, but are also subject to even greater uncertainty from ESI sensitivity. For all these reasons, we caution that the dimer yield estimated here in only a

best guess meant to fit our kinetic model, and that further investigation is needed with more quantitative methodology.

**4 Conclusions and atmospheric relevance**

Contrary to previous chamber experimental results, we have shown here that the reaction of $\alpha$-pinene with $\mathrm{NO_3}$ can form high mass yields of SOA (>21%, at ~30 ppbv of $\alpha$-pinene) under $\mathrm{RO_2}$ fate branching ratios designed to mimic those of summer nights in the Southeast United States. Much of this SOA originates from the self- and cross-reactions of nitrated

peroxy radicals ($n\mathrm{RO_2} + n\mathrm{RO_2}$); we estimate that this pathway alone has an SOA mass yield of 56%, with an associated dimer branching ratio of ~16%, while SOA formation from other $n\mathrm{RO_2}$ pathways is negligible within uncertainty. This hypothesis is also supported by the dominance of dimers in the ozonolysis-subtracted mass spectra of SOA collected from 'simulated nighttime' experiments, the diversity of which suggests that many different peroxy radical rearrangements and dimerization channels contribute to SOA formation from $n\mathrm{RO_2} + \mathrm{RO_2}$ chemistry.

The magnitude of $\alpha$-pinene concentrations in the Southeast United States will be lower than that used in our chamber; thus, we model the $\alpha$-pinene + $\mathrm{NO_3}$ chemistry at nighttime concentrations measured during the SOAS campaign in rural Alabama in the summer (1 ppb $\alpha$-pinene, 1 ppb other terpenes, 20 ppb ozone, 1 ppt $\mathrm{NO_3}$, and 2 ppt $\mathrm{HO_2}$; Ayres et al., 2015; Romer et al., 2018) and assuming steady-state for $n\mathrm{RO_2}$. At atmospheric concentrations, our kinetic model predicts that 80% of $\alpha$-pinene reacts with $\mathrm{NO_3}$. Of the $n\mathrm{RO_2}$ formed, approximately 20% is predicted to react with $\alpha$-pinene-derived $n\mathrm{RO_2}$ or

$_{\mathrm{O3}}\mathrm{RO_2}$ at $k_{\mathrm{RO2+RO2}}$ to $1\times10^{-13}$ cm$^3$ molecule$^{-1}$ s$^{-1}$. Combined with our measured 56% SOA mass yield from $n\mathrm{RO_2} + \mathrm{RO_2}$, this suggests an overall SOA mass yield of 11% from $\alpha$-pinene + $\mathrm{NO_3}$ chemistry under the ambient conditions of SOAS. Combined with our measured 60-70% contribution of organonitrates to the SOA composition, this SOA yield corresponds to a 7%

particulate nitrate yield. The mass of the –ONO$_2$ group alone represents 13% of the SOA mass. A rough calculation predicts that the nighttime reaction of 1 ppbv (~ 5.5 μg/m$^3$) α-pinene may produce ~0.56 μg/m$^3$ SOA (~0.38 μg/m$^3$ of which is particulate nitrates) through its reaction with NO$_3$. These calculations neglect contributions from cross reactions between $n$RO$_2$ and other terpene RO$_2$, estimated to comprise another 20% of the $n$RO$_2$ fate, which may also lead to SOA formation. Our model excludes the contribution of other non-terpene RO$_2$ from precursors such as isoprene, which, while minor at night, may still contribute to RO$_2$ reactivity, and will tend to depress SOA yields due to the smaller dimers formed in those reactions (McFiggans et al., 2019). Although the yields in this work would predict α-pinene + NO$_3$ represents a significant fraction of aerosol and particulate nitrate mass observations during SOAS, previous modeling efforts did not track the SOA or organic nitrates from α-pinene + NO$_3$ due to belief that this reaction does not produce aerosol (Pye et al., 2015; Ayres et al., 2015). Atmospheric models should therefore include SOA formation from α-pinene + NO$_3$, ideally with a dependence on $n$RO$_2$ fate.

Since HRMS analysis suggests that most compounds in α-pinene + NO$_3$ SOA retain their organonitrate functionality, they will become a permanent sink of NO$_x$ following particle deposition. The α-pinene + NO$_3$ chemistry can also have important implications for reactive nitrogen budgets in the gas phase. The high yield of PNP (58%) from $n$RO$_2$ + HO$_2$ will temporary sequester NO$_y$, which may become a permanent sink when the PNP undergoes deposition (Nguyen et al., 2015) or retains its nitrate functionality following further oxidation. Approximately 45% of $n$RO$_2$ is expected to react with HO$_2$ during the conditions of SOAS; thus, the nighttime reaction of 1 ppbv α-pinene is expected to produce ~260 pptv of volatile PNP. During SOAS, PNP is observed with maximum concentrations of ~30 pptv, possibly limited by its atmospheric deposition and transport, among other fates (Nguyen et al., 2015). Together, both particle and gas-phase organonitrate formation and deposition from α-pinene + NO$_3$ chemistry may amount to a substantial removal pathway for atmospheric NO$_x$.

Our results also highlight the necessity of performing chamber experiments under conditions that more closely match those in the ambient atmosphere. Measured SOA yields depend heavily on the reactive fate of RO$_2$ in chamber experiments. Previous experiments using N$_2$O$_5$ as a source of NO$_3$ to study SOA formation from α-pinene + NO$_3$ likely observed low mass yields due to the dominance of $n$RO$_2$ + NO$_3$ reactions, from which we observe little-to-no SOA formation. Similarly, experiments that maximize $n$RO$_2$ + HO$_2$ may observe primarily volatile products. Only in experiments designed to allow sufficient $n$RO$_2$ + RO$_2$ reactions was SOA formation observed. Furthermore, HRMS analysis shows that a large fraction of the compounds in SOA formed under 'simulated nighttime' conditions are not observed in either ozonolysis-only or $n$RO$_2$ + $n$RO$_2$-only SOA, suggesting that synergistic reaction pathways can enhance SOA formation when multiple oxidation channels occur simultaneously. Because ozonolysis and NO$_3$ oxidation occur together at night in the atmosphere, chamber experiments isolating one or the other will necessarily be biased in their representation of nighttime SOA from α-pinene and potentially other volatile hydrocarbons.

*Code and data availability.* Chamber experiment data are available online at the Index of Chamber Atmospheric Research in the United States (ICARUS; https://icarus.ucdavis.edu/). The chemical mechanism for kinetic modelling can be found in the Supplement (Mech. S1).

*Author contributions.* T. B. N., K. H. B. and G. J. P. B. designed the chamber experiments. K. H. B. and G. J. P. B. performed the chamber experiments. K. H. B. performed the kinetic modelling. J. D. C. assisted with chemical syntheses and experiments. T. B. N. and K. H. B. prepared the manuscript.

*Competing interests.* The authors declare that they have no conflict of interest.

*Acknowledgements.* This work was funded by the US National Oceanic and Atmospheric Administration.

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
