# Peer review of "Secondary organic aerosol and organic nitrogen yields from the nitrate radical (NO3) oxidation of alpha-pinene from various RO2 fates"

_Atmospheric Chemistry and Physics, 2021_

## Author Comment (AC1)

We thank the reviewers and community for their careful consideration of our manuscript and their helpful comments. We have reproduced the comments in their entirety below, and have addressed each question and concern individually. Reviewer and community comments are written in black, our responses are in blue, and new text added to the manuscript is italicized.

REVIEWER 1:

This manuscript investigates the SOA yield from a-pinene + NO3 under different RO2 fates. It is shown that significant amount of SOA is produced from RO2+RO2 reactions, while SOA yield from other RO2 reactions, including RO2+NO, RO2+NO3, and RO2+HO2, is minimal. The presented results based on chamber experiments with extensive chemical characterization are convincing. However, the significance of this finding, including the manuscript title, is exaggerated and not well-supported. It is mainly because the RO2 fate at night in the atmosphere is highly uncertain. Previous studies on a-pinene + NO3 focused on RO2+HO2 pathway, because that was believed to be the dominant fate of RO2 at night. What is missing from this manuscript is the evidence that RO2+RO2 is dominant at night. The authors cite Aryes et al. (2015) and Romer et al. (2018) to argue that 30-50% RO2+HO2 and 30-50% RO2+RO2 (Line 225) in the summertime in the SE US. However, the reader glimpsed those references and did not find explicit and careful analysis on the RO2 fate. Another important caveat is that in chamber experiment, a-pinene RO2 reacts with RO2 with 8-10 carbon atoms. In the ambient, however, a-pinene RO2 is more likely to react with RO2 with fewer carbon atoms (i.e., from isoprene), leading to lower SOA yield than chamber experiments. Overall, the reader appreciates the authors' careful work to explore the effects of RO2 fate on SOA formation and recommend publication after the following comments are addressed.

Major Comments
1. Concerns regarding the RO2 fate as elaborated above.

We understand the reviewer's concerns and have sought to emphasize that the experiments are better considered as providing constraints on the SOA and organonitrate yields from each $RO_2$ pathway, which can then be used to calculate ambient production of these quantities of interest depending on conditions at any given site (or in models). We have amended the manuscript title to clarify this (now "Secondary organic aerosol and organic nitrogen yields from the nitrate radical ($NO_3$) oxidation of alpha-pinene from various $RO_2$ fates"). With regards to the citations to Romer et al. (2018) and Ayres et al. (2015), they provided measurements of the species needed to make our own estimates of $RO_2$ fate, as calculated in Section 4; we have rearranged the statement referenced by the reviewer on Line 225 to clarify that the percentages listed there refer to the $RO_2$ fates in our own experiments, and to point the reader to Section 4 for a more detailed discussion of the ambient $RO_2$ fate: "These 'simulated nighttime' experiments (e.g. Experiment 20, **Fig. 1**, right) are termed as such because they provide an atmospherically relevant balance of reactive pathways (60–80% α-pinene + $NO_3$, 20–40% α-pinene + $O_3$; 30–50% $nRO_2$ + $HO_2$, 30–50% $nRO_2$ + $RO_2$) *comparable to those on summer nights in the Southeast United States (based on measurements reported in Ayres et al., 2015 and Romer et al., 2018; see Section 4)*.". With regards to the contribution of $RO_2$ radicals from other precursors to the $RO_2$ + $RO_2$ fate, we have added some discussion of this point to Section 3.3, which is written out in more detail in the response to Reviewer 1's point #5 below. Briefly, isoprene is not emitted or

expected to contribute significantly to the $RO_2$ pool at night, and while other terpenes are likely present, $\alpha$-pinene can represent about half of the $NO_3$ reactivity (and therefore likely half of the $RO_2$ radical pool) at night in the Southeast United States (Ayres et al., 2015).

2. The SOA formation from a-pinene + NO3 is estimated by subtracting the SOA from a-pinene + O3 and a-pinene + OH. Such estimate has severe uncertainty, because of the synergistic reaction pathways, which the authors acknowledges, but did not carefully take into account. The estimated SOA yield from a-pinene + NO3 likely represents an upper limit.

We agree that the subtraction of the ozonolysis-derived SOA fraction has the potential to introduce uncertainty, as acknowledged in the original manuscript, and have expanded on our discussion of the ozonolysis controls in the SI. We have added a table (Table S1) with parameters and measured yields for the ozonolysis controls experiments and a paragraph describing their design. The results in that table show, as many previous experimental efforts have before, that $\alpha$-pinene ozonolysis SOA formation is highly reproducible; we measured a mean SOA yield of 20.4 ($\pm$4.4)% over 20 control experiments with different initial conditions. Additionally, ozonolysis represents only a minor fraction of the reactive fate of $\alpha$-pinene in our simulated nighttime experiments, which inherently limits its contribution to the overall SOA yield. Therefore, the subtraction process itself (subtracting a small number with high precision) does not introduce substantial uncertainty.

The reviewer raises an important point, though, that the identification of synergistic reaction pathways introduces another potential source of uncertainty. However, we view the occurrence of these synergies in our experiments as a benefit, because they are quite likely to occur in the atmosphere; the resulting coefficient estimates from our regression analysis are therefore more representative of each pathway's contribution under ambient conditions than yield estimates that might be derived from chamber experiments isolating each reactive pathway alone. Additionally, we disagree that the estimated yield represents an upper limit; in fact, our experiments with high $nRO_2 + nRO_2$ chemistry and no ozonolysis at all (experiments #12, 14, and 26) had three of the four highest SOA yields we measured (higher than our regression model would predict). This suggests that dimerization of $nRO_2$ with other $nRO_2$ may produce even more low-volatility products that dimerization between $nRO_2$ and $_{O3}RO_2$, which may intuitively be expected given then tendency for $nRO_2$ to contain bulky nitrooxy groups. Additionally, ozone represents a higher fraction of the pinene reactivity in the control experiment compared to the simulated nighttime experiment, which means we are subtracting a higher portion of ozonolysis SOA than occurred in the experiment. Thus, the SOA yield from the nighttime experiment may potentially be considered more of a lower limit than upper. However, we do not have sufficient quantitative data from filter analyses to make these distinctions. We have added a new paragraph to Section 3.1 explaining all of these points:

*"By inherently treating individual reactive pathways as independent variables, these regression analyses cannot separate the possible contributions of interactions between multiple pathways, e.g. from synergistic dimerization between $nRO_2$ and ozonolysis-derived $RO_2$ (see **Section 3.3**). The reported coefficients may therefore misrepresent what each pathway on its own would contribute to SOA formation without such synergy. However, because the analysis was performed on experiments predominantly designed to replicate the reactive pathways*

*experienced by a-pinene under ambient nighttime conditions, we expect the coefficients reported above to represent reasonable estimates of each pathway's contribution to SOA formation in the nighttime atmosphere, including any synergistic reactive pathways that occur in ambient conditions. Conditions which deviate substantially from the nighttime atmosphere and therefore lack the same synergistic reactions (e.g. experiments isolating single reactive pathways) may measure different SOA yields. Among the experiments performed here, those with high $nRO_2$ + $RO_2$ contributions without any ozonolysis (Experiments 12, 14, and 26) exhibited some of the highest measured SOA yields – higher than the regression model would predict – suggesting perhaps that the $nRO_2$ + $nRO_2$ pathway on its own results in even higher SOA yields while $nRO_2$ + other $RO_2$ pathways have lower yields. Without knowing the relative rates of various $nRO_2$ + $RO_2$ reactions, though, we cannot sufficiently constrain these differences, and additional regression analyses including interaction terms between the reactive pathways did not yield statistically robust results."*

3. The estimate of RO2 fate heavily relies on kinetic model, which bears uncertainties in the kinetics of RO2 reactions. The RO2+RO2 rate applied in this study (1e-13) is slower than those in recent findings which report the a-pinene+O3 RO2+RO2 rate is on the order of 1e-12(1) or even 1e-11(2). Although the reasoning for using 1e-13 is briefly mentioned in the manuscript, sensitivity tests regarding the effects of RO2+RO2 rate on the yields of SOA and other products should be conducted.

The brief reasoning described in the manuscript, to which the reviewer refers, did come from sensitivity simulations conducted alongside those we report. In an effort to keep our description concise, we opted to only briefly state in the manuscript that the lack of SOA formation in $nRO_2$+$NO_3$-dominated experiments can only be explained by a $k_{RO2+NO3}$ at least an order of magnitude higher than $k_{RO2+RO2}$, with the best fit in the regression analysis of pathway-specific SOA requiring a difference of two orders of magnitude (this ensures that the $nRO_2$+$NO_3$-dominated experiment, which resulted in very low SOA yields of 3%, had <5% $nRO_2$+$nRO_2$ contribution). We expand upon this description, and the other conclusions of our sensitivity simulations, below and in the main manuscript.

While the $k_{RO2+NO3}$ rate for a-pinene-derived $nRO_2$ is uncertain, the rate we chose ($1.0\times10^{-11}$ $cm^3$ molecule$^{-1}$ s$^{-1}$) is likely an upper limit. Previous studies of $RO_2$+$NO_3$ reactions with non-acyl, non-halogenated peroxy radicals have measured rate coefficients of $(1.1-2.4)\times10^{-12}$ $cm^3$ molecule$^{-1}$ s$^{-1}$ with only a small dependence on molecule size (Orlando and Tyndall, 2012). It seems implausible that the $\alpha$-pinene $nRO_2$+$NO_3$ rate coefficient would be significantly faster than $1.0\times10^{-11}$ $cm^3$ molecule$^{-1}$ s$^{-1}$, but we acknowledge that $RO_2$+$NO_3$ rate coefficients in general are poorly constrained. Our other constraint on $k_{RO2+RO2}$ comes from the contribution of $RO_2$ + $HO_2$ chemistry and our measured PNP yields. $RO_2$ + $HO_2$ rates are better constrained; we use the parameterization in Wennberg et al. (2018), which gives $1.7\times10^{-11}$ $cm^3$ molecule$^{-1}$ s$^{-1}$, similar to the measured value for pinene-derived radicals of $2.1\times10^{-11}$ $cm^3$ molecule$^{-1}$ s$^{-1}$ from Boyd et al. (2003). Increasing $k_{RO2+RO2}$ in our simulations causes a decrease in the contribution of $RO_2$ + $HO_2$ chemistry, but this decrease is only plausible until the measured PNP yield is 100%, which occurs at a $k_{RO2+RO2}$ value slightly above $1\times10^{-12}$ $cm^3$ molecule$^{-1}$ s$^{-1}$. The uncertainty of our CIMS measurements allows a 30% error bound on this maximum, but would still render a $k_{RO2+RO2}$ of $1\times10^{-11}$ $cm^3$ molecule$^{-1}$ s$^{-1}$ implausible.

We also performed the same $RO_2$ pathway SOA regression analysis from the main manuscript on our sensitivity simulation results. We find that scaling the bulk $k_{RO2+RO2}$ rate coefficient down to $1\times10^{-14}$ cm$^3$ molecule$^{-1}$ s$^{-1}$ would result in an estimated SOA yield from $nRO_2 + RO_2$ chemistry 50% higher than what we report, while scaling the bulk $k_{RO2+RO2}$ rate coefficient up to $1\times10^{-12}$ cm$^3$ molecule$^{-1}$ s$^{-1}$ results in an estimated SOA yield from $nRO_2 + RO_2$ 14% lower than what we report in the manuscript for $k_{RO2+RO2} = 1\times10^{-13}$ cm$^3$ molecule$^{-1}$ s$^{-1}$. We have now included these findings, along with more description of our constraints, in two paragraphs (expanded, with new material in italics, from the brief discussion in the initial manuscript) in Section 3.1:

"These regression analyses also intrinsically depend on kinetic model parameters such as bimolecular $RO_2$ reaction rates, some of which are uncertain. While we are unable to fully quantify these rates, we find that certain ratios between rates are constrained by our experimental outcomes. For example, the negligible SOA yield in the high-$NO_3$ Experiment 13 suggests that $nRO_2 + RO_2$ chemistry cannot play a major role in that experiment. This can only be achieved if the bulk rate coefficient for $nRO_2 + RO_2$ reactions ($k_{RO2+RO2}$) is approximately two orders of magnitude slower than that of $nRO_2 + NO_3$ reactions ($k_{RO2+NO3}$). We therefore set the bulk $k_{RO2+RO2}$ to $1\times10^{-13}$ cm$^3$ molecule$^{-1}$ s$^{-1}$ with the corresponding bulk $k_{RO2+NO3}$ equal to $1\times10^{-11}$ cm$^3$ molecule$^{-1}$ s$^{-1}$ *(a value higher than most reported $k_{RO2+NO3}$, although few measurements exist for these reactions). The PNP yields in 'simulated nighttime' experiments also provide a constraint on the relative rates of the $nRO_2 + RO_2$ and $nRO_2 + HO_2$ reactions. $RO_2 + HO_2$ rate coefficients are, in general, are better constrained than those of $RO_2 + RO_2$ or $RO_2 + NO_3$ reactions; we use the parameterization in Wennberg et al. (2018), which gives $1.7\times10^{-11}$ cm$^3$ molecule$^{-1}$ s$^{-1}$, similar to the measured value for pinene-derived radicals of $2.1\times10^{-11}$ cm$^3$ molecule$^{-1}$ s$^{-1}$ from Boyd et al. (2003). Increasing $k_{RO2+RO2}$ in our simulations causes a decrease in the contribution of $RO_2 + HO_2$ chemistry, but this decrease is only plausible until the measured PNP yield is 100%, which occurs at a $k_{RO2+RO2}$ value slightly above $1\times10^{-12}$ cm$^3$ molecule$^{-1}$ s$^{-1}$. The substantial yield of pinonaldehyde instead of PNP from $nRO_2 + HO_2$ calculated by Kurtén et al. (2017) suggests that $k_{RO2+RO2}$ must be well below $1\times10^{-12}$ cm$^3$ molecule$^{-1}$ s$^{-1}$ to keep the PNP yield well below 100%.*

*Based on these constraints, a value of $1\times10^{-13}$ cm$^3$ molecule$^{-1}$ s$^{-1}$ represents our optimal fit and a value of $1\times10^{-12}$ cm$^3$ molecule$^{-1}$ s$^{-1}$ represents an upper limit for the bulk $k_{RO2+RO2}$ of a-pinene $nRO_2$. A higher value would require implausibly high $k_{RO2+NO3}$ values in order to maintain dominance of the $nRO_2 + NO_3$ pathway in Experiment 13, and would require implausibly high $k_{RO2+HO2}$ values in order to maintain a measured PNP branching ratio below the maximum of 100%. Our best estimate is reasonably within the broad range of measured $k_{RO2+RO2}$ for other peroxy radicals, particularly given that rate coefficients tend to decrease with size and degree of substitution at the radical site (Orlando and Tyndall, 2012). It is smaller, however, than some recent measurements of the $k_{RO2+RO2}$ of peroxy radicals derived from α-pinene ozonolysis ($1\times10^{-12}$ to $1\times10^{-11}$ cm$^3$ molecule$^{-1}$ s$^{-1}$), highlighting the uncertainty in reaction rates of this type (Zhao et al., 2018; Berndt et al., 2018b). A slower bulk $k_{RO2+RO2}$ of $1\times10^{-14}$ cm$^3$ molecule$^{-1}$ s$^{-1}$, which would still be consistent with our constraints on relative rate ratios,* would scale the contribution of $nRO_2 + RO_2$ chemistry in the 'simulated nighttime' experiments by a factor of 0.67, which in turn would require scaling SOA yields from this pathway up by 50%. *A faster bulk $k_{RO2+RO2}$ of $1\times10^{-12}$ cm$^3$ molecule$^{-1}$ s$^{-1}$ would scale the $nRO_2 + RO_2$ contribution in the same experiments by a factor of 1.16, which in turn would require scaling the $nRO_2 + RO_2$ SOA yield down by 14%.*

*Despite our constraints, the bulk $k_{RO2+RO2}$ remains highly uncertain, and is also likely to differ between nRO$_2$ isomers for self- and cross-reactions (Orlando and Tyndall, 2012)."*

4. Because of the unclear RO2 fate at night, the manuscript title is not appropriate. The reader suggests something like "SOA yield from a-pinene + NO3 under different RO2 fate"

We have changed the manuscript title to "Secondary organic aerosol and organic nitrogen yields from the nitrate radical (NO$_3$) oxidation of alpha-pinene from various RO$_2$ fates"

5. When simulating the RO2 fate under summertime conditions observed in the SE US, the kinetic model only contains a-pinene chemistry. As a-pinene only accounts for a very small fraction of VOC in the ambient, the ambient RO2 fate is largely driven by the chemistry of other VOCs. In other words, the simulated RO2 fate does not represent the RO2 fate in the SE US. This challenges the representativeness of the "simulated nighttime experiment" and should be carefully acknowledged in the manuscript.

Although α-pinene makes up a small fraction of VOC in the ambient, its fractional contribution to nighttime RO$_2$ chemistry is amplified due to (a) the low amplitude of the diurnal cycle of its emission rate relative to that of isoprene and other compounds emitted exclusively during the day, and (b) its rapid reaction with NO$_3$ relative to other VOCs. Thus, as Ayres et al. (2015) show, α-pinene can contribute half of the nighttime NO$_3$ reactivity (and, accordingly, approximately half of RO$_2$ radicals at night) in the Southeast United States in the summer.

We agree, however, that the contribution of these other RO$_2$ + RO$_2$ pathways can be important. We have already included reactions of other terpenes in our modeling (Section 4) of the α-pinene fate in at night in the summertime Southeast United States, and described their contribution in the manuscript: "These calculations neglect *contributions from* cross reactions *between nRO$_2$ and* other terpene RO$_2$, estimated to comprise another 20% of the nRO$_2$ fate, which may also lead to SOA formation." We have also added a sentence describing the potential importance of other RO$_2$ + RO$_2$ pathways: "*Our model excludes the contribution of other non-terpene RO$_2$ from precursors such as isoprene, which, while minor at night, may still contribute to RO$_2$ reactivity, and will tend to depress SOA yields due to the smaller dimers formed in those reactions (McFiggans et al., 2019).*"

Minor Comments
1. Line 240. Why is the SOA yield higher in RO2+NO than RO2+NO3?

We did not examine the RO$_2$ + NO pathway in great detail, as it is not expected to contribute appreciable to the ambient fate of a-pinene nRO$_2$. The multivariate linear regression estimate for the SOA yield from the RO$_2$ + NO pathway thus relies on only three experiments, and therefore comes with a high uncertainty (11(±11)%). As we write in Section 3.1, "*SOA mass yields from the other pathways* [aside from nRO$_2$ + nRO$_2$] *are not significantly different from zero within uncertainty*". We cannot say, with these uncertainty bounds, that the SOA yield from the RO$_2$ + NO pathway is "higher" than that of RO$_2$ + NO$_3$, which is why we do not discuss the relative values of these yields further. We can speculate that differences in the branching ratio to

organonitrate formation or in the leftover energy available for intramolecular rearrangement following alkoxy radical formation between the $RO_2 + NO$ and $RO_2 + NO_3$ pathways may be responsible for their differing SOA yields, but we do not have experimental evidence to investigate these disparities further.

2. In figure 4, all experiments seem to fall into two groups, six experiments above the dashed line and the majority of experiments below the line. Does any factor drive the segregation? Labeling the data points by experiment would be useful and a good start point. What's the regression slope (i.e., ozonolysis-corrected SOA yield) if only experiments below the dashed line are fitted?

Figure 4a, which exhibits the grouping to which the reviewer refers, is by design only able to show the correlation between the $nRO_2 + nRO_2$ reactive fraction and the SOA yield – it is not meant to provide a perfect fit. The deviation from this line is, therefore, related to any factors other than the $nRO_2 + nRO_2$ reactive fraction that might influence the SOA yield. This includes the contributions from other reactive pathways, which are accounted for in Figure 4b, which explains why the fit in 4b is superior to that in 4a.

We have made a number of edits to the manuscript that hopefully make this point clearer and provide some of the analysis in which the reviewer expresses interest. First, we have restricted our regressions on SOA yields to seeded experiments, which removes two of the outlier points. The new regression coefficient for the $nRO_2 + nRO_2$ fraction is 58(±6)% (vs. the previous coefficient of 67(±7)%), and the coefficient excluding the remaining four points above the dashed line to which the reviewer refers is 52(±4)%. Because these coefficients are not significantly different, and no other experimental considerations suggest that the points above the dashed line should be excluded from the regression analysis, we opt to keep them in. Second, we have reduced Figure 4 to one panel, which shows the regression results from Figure 4b with point now colored by the $nRO_2 + nRO_2$ fraction [($\alpha$-pinene+$NO_3$+$RO_2$) / ($\alpha$-pinene reacted)], helping to highlight the dominant factor in the regression without implying that the $nRO_2 + nRO_2$ fraction should alone be able to explain the SOA yields. Third, we have relocated panel 4a to an SI figure which also shows correlations with other pathways, expanded this figure to also show dependences on the ozonolysis and OH pathway contributions, and labeled the points as requested. Finally, we have added an addition SI figure replicating panel 4b and coloring the points by additional experimental variables (seed aerosol loading, $[O_3]_0$, $[H_2O_2]_0$, $[\alpha$-pinene$]_0$), which shows that these other variables are unable to explain the remaining spread in the measured SOA yields.

REVIEWER 2:

Major comments:

In the results section, it is mentioned there are effectively 5 characteristic experiments that were performed: RO2 + NO3 (Fig 1 left), RO2 + RO2 (Figure 1 middle), RO2 + NO (not shown), RO2 + HO2 (not shown) and "simulated night time" (Figure 1 right). The two that are not shown should be included for comparison sake in the supplement.

The RO$_2$ + NO case was left out of Fig. 1 because it was considered inconsequential, as that pathway contributes little to the $n$RO$_2$ fate at night in the Southeast United States. We agree that it might be instructive for some readers and nice for completeness' sake, though, so we have added an analogous time trace plot for Experiment 15 (high-NO) to the Supplement in Figure S1. As for the RO$_2$ + HO$_2$ case, we explain in the second paragraph of Section 3.1 that it cannot be isolated in the same way as the other pathways; we therefore do not have a "characteristic" RO$_2$ + HO$_2$ experiment to show. Instead, we use varying initial conditions in the "simulated nighttime" experiments to emphasize the relative contributions of the RO$_2$ + HO$_2$ and RO$_2$ + RO$_2$ pathways. To show that varying these conditions can indeed result in different contributions of the two reactive pathways, we have also added time traces from Experiments 7 & 8 to Figure S1.

Line 210: I understand that there must be careful caveats to do this subtraction shown in Figure 2. But if there were control experiments that were performed for each of the experiments listed (see line 122) the information for the control experiments should be added. Also, the SOA yield in Table 2 should indicate if this is the subtracted / corrected yield for NO3 chemistry or the total yield (with O3 contribution included). Where applicable the control experiment SOA yield needs to be included.

Please see the response to Reviewer 1's major concern #2 for a more detailed discussion of the changes we've made to our description of the ozonolysis controls and the yield correction process. We hope these edits have helped to clarify the role of the ozonolysis-only controls. In addition, we have added a footnote to Table 2 indicating that the reported yields are ozonolysis-corrected, and have added a detailed table to the SI (Table S1) with ozonolysis control experiment parameters, measured SOA yields, and correction factors.

Along with these considerations, the authors could consider performing a sensitivity analysis when comparing SOA yields to the fate of RO2 + RO2. On this note, I am surprised it looks like there is no value on Figure 4b that is 0 on the y-axis, all appear to be positive.

We agree with the reviewer that such sensitivity analysis may prove beneficial, and have now performed sensitivity analyses with regards both to the uncertainties in RO$_2$ reaction rates in the kinetic model and to the regression analysis of the SOA yields. Please see the responses to Reviewer 1's major comment #3 and minor comment #2, respectively, for more detail on these sensitivity analyses. With regards to Figure 4b, we indeed measured above-zero SOA yields (before ozonolysis correction) in all experiments except the unseeded nucleation experiment (which is excluded from the yield vs. RO$_2$ fate analysis).

Line 322: It should be noted, that regardless of seed concentration, wall loss will still affect SOA yields. See DOI: 10.1021/acs.est.0c03381

The cited study by Krechmer et al. (2020) largely focused on continuous flow reactors. They even state "we hypothesize that faster [oxidant] injection [in batch mode reactors] would result in a higher condensation sink earlier in the experiment and decrease the effect of gas-phase wall losses relative to the CF experiment", and that the lack of stirring in batch mode decreases the wall loss sink. Furthermore, the lack of any observable dependence of SOA yield on seed surface

area above ~150 $\mu m^2$ $cm^{-3}$ supports our claim that vapor wall losses have negligible effects at the high loadings used in our $RO_2$ fate experiments; even for the SVOCs that Krechmer et al. postulate establish gas-liquid equilibria on the walls and particles (which may not apply in our dry conditions), some seed area (or volume) dependence is expected. We therefore believe that the high seed conditions used in our $RO_2$ fate experiments have effectively minimized vapor wall loss effects to a point that they do not contribute substantially to measurement uncertainty. However, we acknowledge that such effects are important to consider, and have added a reference to it with a citation to Krechmer et al. in Section 3.2 of the manuscript: "*Although vapor wall losses may still cause a decrease in measured SOA yields regardless of seed surface area (Krechmer et al., 2020), we assume here that these effects are minimal at the high seed surface areas (> 600 $\mu m^2$ $cm^{-3}$) used in our experiments to investigate $RO_2$ fate. However, true SOA yields may be higher than reported if vapor wall losses are considered.*"

Line 339: It appears that there is a disconnect between the lack of nucleation and the assertion that apinene + NO3 produces lower volatility molecules, considering nucleation is connected to the formation of low-volatility or extra low-volatility molecules. Why would apinene + NO3 not nucleate (as discussed in the experiments here), while Toluene SOA from Zhang et al (2014) does? (note the 0 seed surface area is not zero in Zhang et al.) I think more caution is required in the authors statement.

We agree with the reviewer that the assertion that α-pinene SOA volatility in our experiments is lower than in Zhang et al. (2014) is not fully supported by our results, as differences in the environmental chambers themselves could play a major role. We have thus removed that comment ("suggesting much lower volatility for the α-pinene + $NO_3$ products"). Nucleation will depend on a number of factors, including the volatility of oxidation products but also the rate of oxidation and precursor concentration in the experiment, both of which influence the concentrations of low-volatility products during the experiment. In comparison to Zhang et al. (2014), in which a different VOC precursor reacts with a different oxidant in a different chamber, we cannot disentangle these effects relative to our own experiments. It may be that the higher toluene concentrations used by Zhang et al. (2014) relative to the α-pinene concentrations in the present study increased the concentrations of low-volatility products enough to initiate nucleation; alternatively (or additionally), it may be that the toluene SOA in Zhang et al. (2014) is even lower-volatility than the α-pinene SOA formed here, as has been shown previously in comparisons of SOA from these two precursors (see Kim et al., 2013; DOI: 10.5194/acp-13-7711-2013). Regardless, we have added further caveats to the relevant statement in the manuscript: "No SOA formation was observed in the absence of seed aerosol, indicating that α-pinene + $NO_3$ *with simulated nighttime $RO_2$ fates* does not nucleate *on its own at the low precursor concentrations used in these experiments (and the even lower concentrations present in the atmosphere).*"

Line 353: It is said that 34% of the peaks are found to come from ozonolysis experiments, how much does this contribute to the overall intensity. It may be good to discuss this in terms of number of peaks and how important is their relative contribution.

We apologize for the confusion; this analysis was performed in terms of peak intensity (or signal), rather than number of peaks. This is a better metric for the contribution of each pathway

to the overall particle mass than number of peaks, as the vast majority of discernible peaks contribute very little to the total signal. We have clarified throughout this paragraph.

Line 374-376: The other option to form trimers would be through the RO2's in Table 2, that combine to form dimers and possess a C=C. This would be nRO2(e and f). These could undergo another reaction with $NO_3$ to form a C20 RO2, which would react with C10 RO2 radicals to form C30s.

We thank the reviewer for this excellent point; we have added a mention of this alternative pathway to the manuscript: "… *or may alternatively be attributable to second-generation accretion reactions of dimers (especially those still containing a double bond, which would react rapidly with $NO_3$) with $RO_2$ monomers*."

Paragraph (line 398): So the RO2 + RO2 → dimer branching ratio is based on the dimers measured by the offline measurement. On line 175, the authors state that the HRMS data is qualitative. There is a disconnect between the statement in the methodology and the use of the dimers to constrain the branching ratio of RO2 + RO2. Could the authors talk about error associated with this measurements and how certain they are of the 40-60% reported?

We agree with the reviewer that the 18% dimer yield from $nRO_2 + nRO_2$ based on the HRMS data may depend on the quantifiability of that data. We have therefore removed mention of the 18% dimer yield from the abstract, and added some discussion of caveats on the measured yield to the end of Section 3.3: "*However, our estimated dimer yield is highly uncertain for a wide range of reasons, including (a) variations in compound-specific sensitivities within the ESI HRMS, which may accentuate the dimer contribution as ESI efficiency has been shown to increase with molecular size (Kenseth et al., 2020); (b) sensitivity to the uncertain estimates of $RO_2$ fate, which depend on poorly-constrained $RO_2 + RO_2$ and $RO_2 + NO_3$ rates (see Section 3.1); (c) the possibility of particle-phase reactions, or reactions during filter extraction, which may form or break dimers; and (d) lack of consideration of trimers, which account for 14% of overall peak intensity and may form from dimers, but are also subject to even greater uncertainty from ESI sensitivity. For all these reasons, we caution that the dimer yield estimated here in only a best guess meant to fit our kinetic model, and that further investigation is needed with more quantitative methodology.*"

Why are the trimers not included in the contribution for RO2 + RO2? The authors postulate that these molecules could form from C20 dimers.

As noted above, we are now avoiding detailed quantification of these pathways from the HRMS data, and have therefore opted not to recalculate these values with the trimers included. While the total trimer signal contributes 14% of the overall peak intensity (across positive and negative mode combined), we expect that the trimers will have an even higher sensitivity bias than the dimers with the electrospray ionization method used, and therefore their contribution would likely be overestimated. We have added mention of the trimer contribution (and its associated uncertainty) to the end of Section 3.3 (see response to previous comment).

Also, on line 351, the peaks that are observed in the RO2 + RO2 experiments make up 29% (positive mode) and 39% (negative mode) of the simulated night time experiment. Wouldn't a lower limit of 29% be more appropriate? With the given range, how sensitive is the RO2+RO2 → dimer branching ratio to the dimer fraction? With the change of the branching to RO2 + RO2, how were the other RO2 + RO2 branching ratios altered? Is the model sensitive to these pathways? (or is it consistent with your findings above)

The 29% fraction does not come into play in these estimates of dimer formation, as we use the dimer fraction from both $nRO_2 + nRO_2$ and synergistic $nRO_2 + {}_{OH/O_3}RO_2$ to constrain the dimer branching. The 29% and 39% fractions represent peaks unique to the $nRO_2 + nRO_2$ experiment, but not those observed in both pathways or those from synergistic pathways. Furthermore, the positive and negative mode signals more closely represent two separate populations of dimers observed in the SOA, rather than two estimates of the same fractional dimer contribution, because the HRMS analysis is sensitive to different species in each mode. Therefore, the true fractional dimer population should include both modes' signals, making an average more appropriate than the use of one mode or the other as a limit.

 Regardless, because we are now reducing the emphasis on yield quantification from this data, we do not go into detail on these sensitivities here. To answer the penultimate question here, we have clarified how branching ratios are altered by adding the following (italicized) statement to the last paragraph of Section 3.1: "we apply the bulk effective yield to all peroxy radical self- and cross-reactions *and scale down the pinonaldehyde-forming pathway accordingly*". Because we do not measure pinonaldehyde in our experiments, we cannot constrain this scaling, but pinonaldehyde remains the dominant simulated product in most of these experiments.

In the discussion on the synergy of the different reaction pathways of RO2 radicals is suggested in many works including: (Zhao et al., 2018;Heinritzi et al., 2020;Berndt et al., 2018a;Berndt et al., 2018b;McFiggans et al., 2019)

We thank the reviewer for pointing this out; our reference to Teng et al. (2017) here was erroneous. We have instead added the references noted by the reviewer in addition to Kenseth et al. (2018) and Inomata (2021).

Minor comments:

Line 74: The reference doesn't appear to match the citation of Kurten et al.

The reference has been updated.

Line 80: reinvestigated

The hyphen has been removed.

Lines 93-94: It could be noteworthy to mention that at elevated humidities, N2O5 uptake into the particle could be an important loss mechanism / source of radicals in the particle phase. Also, at elevated RH there will be some uptake of H2O2, which is useful for your experiments with HO2.

We have added a mention of the "the effects of gas-particle partitioning of $H_2O_2$ and $N_2O_5$" to this sentence.

Line 95: RH not defined.

"Relative humidity (RH)" has been added.

Line 183: minor grammatical comment, do you mean to say something along the lines of: experiment 13 is representative of RO2 + NO3 chemistry? I don't think "easy to isolate" is necessarily the best way to phrase it.

We have replaced "easy to isolate" with "dominant" in describing the role of the $RO_2 + NO_3$ reaction in Experiment 13.

Line 189 minor grammatical comment: a domination? Maybe… achieves a $RO_2 + HO_2$ that dominates the $RO_2$ reactivity.

We have replaced "that can achieve a domination of $nRO_2 + HO_2$ chemistry" with "whereby reaction with $HO_2$ represents a majority of the $nRO_2$ fate".

Figure 4 is mentioned in the text as (a) and (b) However, they are not labeled as such.

(a) and (b) labels have been added to Figures 2 and 5; Figure 4 no longer has two separate panels.

Line 284: I believe there is an "alpha" or "beta" missing in front of pinene.

"α" has been added.

Figure 7 caption. What are the black lines in Figure 7c?

We thank the reviewer for catching this; we mistakenly included a version of Figure 7 in which the negative spectrum of Panel B did not have black peaks, which made their inclusion in Panel C confusing. We have updated the figure, and revised the caption to clarify.

Table 2 and Figure 8: It seems like there is a missing nRO2 from Table 2 that is not included in Figure 8. (nRO2 - f)

We thank the reviewer for noticing this oversight; Figure 8 has been updated.

Line 352: "Negative mode may overestimate the contributions from dinitrates dimers from nRO2 + nRO2" do you mean RO2 + RO2 chemistry specifically, or generally the experiment where the RO2 + RO2 is the dominant radical pathway?

We have added "chemistry" to clarify.

Line 355: Same comment as on Line 352, but just about the usage of RO2 + RO2.

We have added "control experiments" to clarify.

Line 383: same comment as above about nRO2 + nRO2.

We have added "in Experiment 26" to clarify.

Line 425-426: "This SOA yield corresponds to a 9% particulate nitrate yield for compounds that did not hydrolyze." I don't understand how this was determined, I think there is a citation or two missing. Plus, with the experiments conducted under dry conditions, I don't understand how any molecules here would hydrolyze.

The nitrate yield was determined simply from our measured contribution of organonitrates to the total SOA composition signal in Section 3.3; we have added a clause on this sentence clarifying that calculation and have removed the reference to hydrolysis.

COMMUNITY COMMENT:

To improve the accessibility of the suggested mechanism, and for easier comparisons to representations of a-pinene oxidation in other atmospheric chemistry models, I suggest the authors include the SMILES strings that correspond to each compound within the Supplement of the paper. For any lumped molecules, an example of the SMILES string for such a compound that would be sufficient.

SMILES strings for the stable molecules in Mechanism SI have been added to the list of species in the SI.